# Learning Where It Matters: Responsible and Interpretable Text-to-Image Generation with Background Consistency

**Sayedmoslem Shokrolahi**                                                    *20ss184@queensu.ca*
*Department of Electrical and Computer Engineering*
*Queen's University*

**Jae–Mo Kang**                                                              *jmkang@knu.ac.kr*
*Department of Artificial Intelligence*
*Kyungpook National University*

**Il-Min Kim**                                                              *ilmin.kim@queensu.ca*
*Department of Electrical and Computer Engineering*
*Queen's University*

**Reviewed on OpenReview:** *https://openreview.net/forum?id=sCOJGbJwAJ*

## Abstract

Text-to-image diffusion models have achieved remarkable progress, yet they still struggle to produce unbiased and responsible outputs. A promising direction is to manipulate the bottleneck space of the U-Net (the *h*-space), which provides *interpretability* and *controllability*. However, existing methods rely on learning attributes from the entire image, entangling them with spurious features and offering no corrective mechanisms at inference. This uniform reliance leads to poor subject alignment, fairness issues, reduced photorealism, and incoherent backgrounds in scene-specific prompts. To address these challenges, we propose two complementary innovations for training and inference. First, we introduce a spatially focused concept learning framework that disentangles target attributes into concept vectors by suppressing target attribute features within the multi-head cross-attention (MCA) modules and attenuating the encoder output (i.e., *h*-vector) to ensure the concept vector exclusively captures target attribute features. In addition, we introduce a spatially weighted reconstruction loss to emphasize regions relevant to the target attribute. Second, we design an inference-time strategy that improves background consistency by enhancing low-frequency components in the *h*-space. Experiments demonstrate that our approach improves fairness, subject fidelity, and background coherence while preserving visual quality and prompt alignment, outperforming state-of-the-art *h*-space methods. The code is provided at `https://github.com/Moslem-Sh21/learning-where-it-matters`.

## 1 Introduction

Diffusion models (DMs) have emerged as a leading framework for image generation, demonstrating strong performance since the introduction of Denoising Diffusion Probabilistic Models (DDPMs) Ho et al. (2020); Sohl-Dickstein et al. (2015); Song et al. (2020). By leveraging iterative denoising, they produce high-quality, photorealistic images and are easily conditioned on text prompts Rombach et al. (2022); Ramesh et al. (2022); Karras et al. (2022); Peebles & Xie (2023); Balaji et al. (2022); Saharia et al. (2022); Qu et al. (2024); Podell et al. (2024). However, this flexibility introduces challenges, particularly in achieving responsible and unbiased image generation. Issues such as implicit bias, ethical misalignment, and unsafe content highlight the pressing need for methods that guide these models toward responsible outputs Gandikota et al. (2023); Schramowski et al. (2023); Kumari et al. (2023); Gandikota et al. (2024); Li et al. (2024b).

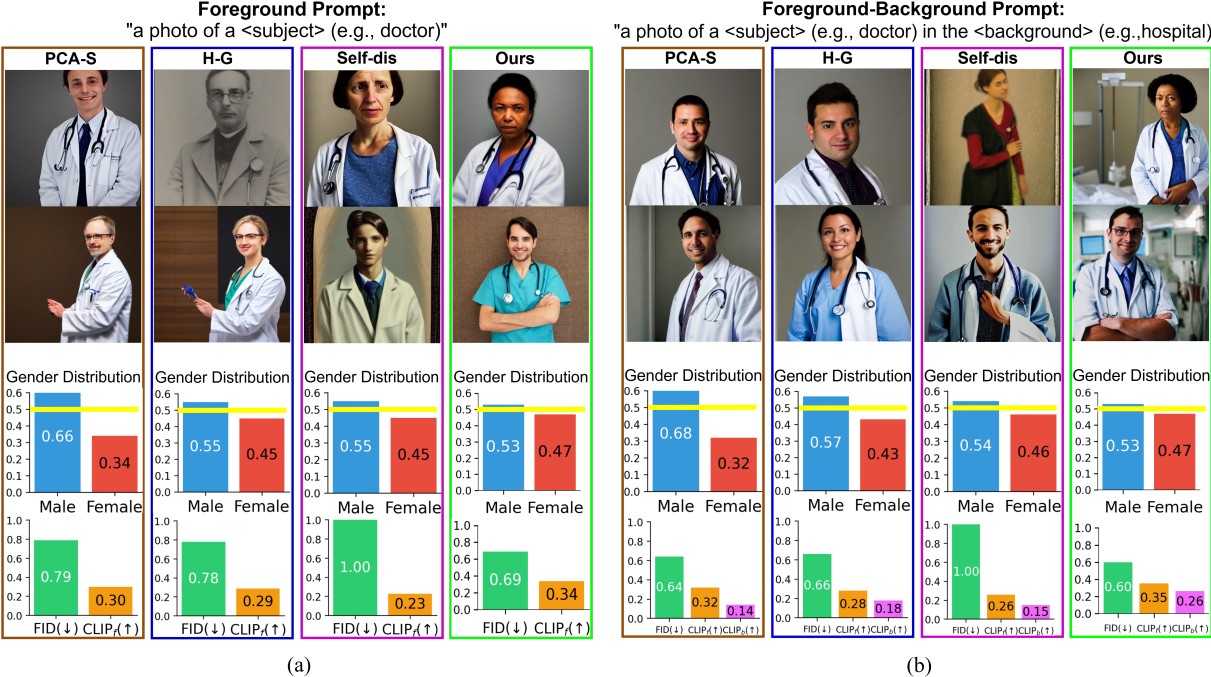

Figure 1: Comparison of fairness, image quality, semantic alignment, and background generation across methods. Metrics are based on 150 generated images per method. Gender distribution histograms assess fairness (yellow line: ideal). FID (scaled by $1/100$) measures image quality. $\text{CLIP}_f$ scores reflect alignment with the subject term (doctor), and $\text{CLIP}_b$ scores (Fig. 1(b) only) reflect alignment with the background term (hospital). (a) Foreground-only prompts: Our method improves image quality, fairness, and subject identity. (b) Foreground-background prompts: Our method generates accurate backgrounds while maintaining fairness and alignment. Extensive results for 36 different prompts (listed in Appendix A) are provided in Table 1. All images across all methods are generated using the same random seed.

Existing methods for fair and safe image generation in DMs can be broadly categorized based on the component of the model they target for intervention. Prompt-based methods aim to mitigate bias by filtering or augmenting the input text, as demonstrated in Chuang et al. (2023); Ni et al. (2024); Brack et al. (2023). Text-encoder-based approaches steer generation by modifying the learned text embeddings Gal et al. (2023); Motamed et al. (2025); Kim et al. (2025). A large body of work focuses on fine-tuning the entire model or selected layers to enforce responsible behavior during generation Bui et al. (2024); Li et al. (2024a); Gandikota et al. (2023); Kumari et al. (2023); Gandikota et al. (2024); Gong et al. (2024); Choi et al. (2023). Another line of research involves editing the representation at the input of the U-Net, enabling concept control through non-linear transformation as in Park et al. (2023); Meng et al. (2021); Tsaban & Passos (2023). Additionally, modifying the noise prediction during reverse diffusion has also been proposed for responsible generation Dalva & Yanardag (2024); Schramowski et al. (2023); Meng et al. (2021).

A particularly promising direction is to manipulate the bottleneck layer space of the U-Net, known as the $h$-space Haas et al. (2024); Li et al. (2024b); Parihar et al. (2024). In text-to-image DMs, the $h$-space represents a semantic latent space that captures representations of specific attributes such as gender or race. By strategically manipulating this space, the image generation process can be steered toward fair and appropriate generated images without retraining the model.

The $h$-space approach offers two main advantages, foremost being *interpretability*. The representations in the $h$-space, referred to as the $h$-vectors, capture distinct semantic attributes (e.g., gender, age), revealing how semantic and visual concepts are encoded Li et al. (2024b); Haas et al. (2024); Parihar et al. (2024). Because they correspond to specific attributes, these vectors can be directly manipulated to influence generation, enabling bias identification and alignment with human expectations. The second advantage of the $h$-space

approach is the *linear controllability*, enabling flexible control over generation Haas et al. (2024); Li et al. (2024b); Parihar et al. (2024). Owing to the linear controllability, the semantically meaningful $h$-vectors can be scaled or combined to adjust concept strength or create attribute mixtures, offering practical benefits for real-world applications. Motivated by these two compelling advantages, we focus on learning target attributes in the $h$-space and leveraging them during inference for responsible image generation.

Existing $h$-space methods typically learn target attributes from the *entire* image region Li et al. (2024b); Parihar et al. (2024); Haas et al. (2024), which can entangle them with spurious attributes. Without spatial distinction, the resulting concept vectors risk capturing mixed attributes, thereby reducing specificity. To overcome this, we propose an innovative spatially focused attribute learning strategy that intelligently suppresses target attribute features in localized regions during the learning step. This localized suppressing approach helps disentangle the target attributes from spurious features and enables more precise control in the $h$-space. Another critical yet overlooked component of prompt-aligned image generation is the inference pipeline. Prior $h$-space methods have ignored this stage entirely, lacking mechanisms to enforce prompt–image consistency. To address this gap, we introduce a novel and highly effective inference-time technique, the first of its kind in $h$-space frameworks, that significantly improves prompt-image alignment. Notably, previous $h$-space methods entirely disregarded both localized learning and inference-time strategies, an oversight that led to four major limitations.

One limitation of existing methods is their occasional failure to generate images that align with the prompt. For example, when the prompt is "a photo of a doctor", the generated image occasionally lacks distinguishing attributes of a doctor. Second, their ability to ensure fairness across different societal groups, such as gender and race, remains suboptimal. Third, these approaches often result in poor image quality, producing outputs that lack photorealism. Fourth, they struggle to accurately generate background content. To address these issues, we propose novel methods in both the concept vector learning and inference steps.

To address the first three foreground-related limitations, we propose a novel spatially focused attribute learning method, in contrast to prior $h$-space approaches that rely on the entire image. Our method learns a *concept vector* in the $h$-space that exclusively captures the target attribute by locally suppressing it within the $h$-space before incorporating the concept vector. Specifically, to ensure that the concept vector serves as the *sole* component for capturing target attribute features, we attenuate its presence in the multi-head cross-attention (MCA) module by masking pixels related to the target attribute using the proposed attribute-separation masks. In parallel, we suppress overlapping target attribute representations in the encoder output ($h$-vector) through spatial attenuation guided by attribute-attentive heatmaps. Finally, we introduce a spatially weighted reconstruction loss that directs optimization toward attribute-relevant regions. Collectively, this triple local modulation strategy enables precise and effective encoding of the target attribute within the concept vector.

Existing $h$-space methods focus mainly on simple *foreground prompts* (e.g., "a photo of a <subject>"), but real-world applications often require *foreground–background prompts* (e.g., "a photo of a <subject> in the <background>"). Notably, existing methods frequently fail to generate accurate backgrounds in such cases. To overcome this, we introduce a novel inference-time technique that enhances low-frequency components in $h$-space. Operating solely during inference, it is compatible with any $h$-space method, enabling more accurate background generation.

Fig. 1 presents a comparative analysis of unbiased image generation across two prompt types: a foreground prompt ("a photo of a doctor", Fig. 1(a)) and a foreground–background prompt ("a photo of a doctor in the hospital", Fig. 1(b)). As illustrated in Fig. 1, existing $h$-space methods exhibit all four major shortcomings—subject misalignment, fairness imbalance, reduced photorealism, and incoherent backgrounds. In contrast, both quantitative and qualitative results demonstrate that our approach substantially improves fairness and visual quality while maintaining subject accuracy and background fidelity. By addressing these critical limitations, our proposed framework remains model-agnostic and can be seamlessly applied to any diffusion model built upon a U-Net architecture. The main contributions of this work are summarized as follows:

- We propose a method for precise concept vector learning in the $h$-space to generate responsible images. Using attribute-separation masks, attribute-attentive heatmaps, and spatially weighted

loss, our approach focuses on target regions, ensuring concept vectors capture attributes accurately. This improves image quality and alignment with the input prompt.

- We introduce a new inference-time generation method that accurately synthesizes both foreground and background content. The core idea is to enhance the low-frequency components in the $h$-space during generation.

- From extensive experiments, we show that our method achieves high-quality and responsible image generation with improved fairness, subject fidelity, and background consistency, specifically targeting to learn *interpretable* and *(linearly) controllable* concept vectors.

## 2 Related Works

Ensuring unbiased text-to-image generation is challenging due to the impracticality of perfectly cleaning large-scale training datasets. Existing mitigation strategies intervene at different stages of the diffusion pipeline, each with trade-offs. Prompt-based methods steer generation by filtering or augmenting input prompts Chuang et al. (2023); Ni et al. (2024); Brack et al. (2023), but cannot fix biases embedded in model representations. Text-encoder interventions introduce learnable embeddings to influence outcomes Gal et al. (2023); Motamed et al. (2025); Kim et al. (2025), yet remain limited in addressing biases within the denoising model parameters.

Model fine-tuning offers a more direct solution by updating network weights, including cross-attention or U-Net layers Bui et al. (2024); Li et al. (2024a); Gandikota et al. (2023); Kumari et al. (2023); Gandikota et al. (2024); Gong et al. (2024); Choi et al. (2023). However, it is computationally intensive and prone to overfitting. Other methods manipulate U-Net inputs Park et al. (2023); Tsaban & Passos (2023); Meng et al. (2021) or modify predicted noise during reverse diffusion Dalva & Yanardag (2024); Schramowski et al. (2023); Meng et al. (2021). However, this approach lacks concept interpretability.

A promising and practically attractive approach for real world deployment is to manipulate the $h$-space. This line of works operates on the bottleneck layer of the U-Net and exploits the interpretability and linear property of the $h$-space Li et al. (2024b); Parihar et al. (2024); Haas et al. (2024). Researchers have demonstrated that semantic attributes such as gender and age can be extracted by applying linear techniques like Principal component Analysis (PCA) in the $h$-space Haas et al. (2024). In Parihar et al. (2024), a linear classifier was trained in $h$-space, but its effectiveness was limited by its reliance on the content of the entire image as true labels. Self-dis Li et al. (2024b) generates images using prompts with the target attribute and reconstructs them from modified prompts without the target attribute, then trains a concept vector in the $h$-space using reconstruction loss based on the entire image content.

Despite their meaningful contributions (and their inherent interpretability and linear controllability), existing $h$-space methods continue to face four key limitations: (i) occasional subject misalignment, (ii) limited fairness across groups, (iii) reduced photorealism, and (iv) poor background generation. These challenges motivate our proposed approach for responsible, high-quality image generation with faithful subject and background content.

## 3 Proposed Method

To address the aforementioned limitations, we propose a novel framework comprising two strategies. First, we introduce an approach that aims to comprehensively and exclusively encode the target attribute into a vector in the $h$-space, concept vector $\mathbf{v}$. To this end, we develop effective mechanisms to suppress target attribute features at the output of encoder, and we also design a new loss function. Second, we develop an inference-time strategy that enhances background generation by amplifying low-frequency components in the $h$-space. Each strategy is detailed in the following subsections.

### 3.1 Concept Vector Learning Through Target Attribute Suppression

Despite recent progress, existing $h$-space methods for fair image generation remain limited because they learn concept vectors from the *entire* image rather than focusing on specific regions where the features for the target attribute are actually encoded. Such strategies inevitably lead to undesirable entanglement between target and spurious attributes, preventing the target attribute from being captured exclusively and comprehensively within the concept vector $\mathbf{v}$. As a result, these methods suffer from issues such as subject misalignment, uneven fairness across groups, and reduced photorealism as shown in Fig. 1.

To illustrate, let us examine the scenario of capturing the target attribute $\mathcal{T}$="female", as shown in Fig. 2. The system consists of a (pre-trained) main DM, denoted by $\mathcal{M}$, along with its duplicate $\mathcal{M}'$, both kept frozen (for notational simplicity, we omit the time step $t$ throughout). The objective is to encode $\mathcal{T}$ in a concept vector $\mathbf{v}$. As shown in Fig. 2(a), the existing approach begins with inputting a *target-included* prompt $\Phi$="a female person" into $\mathcal{M}'$ to generate an image $\mathcal{I}$ containing $\mathcal{T}$, which is then given to $\mathcal{M}$ for the forward diffusion process to learn $\mathcal{T}$. To meet the goal of fully and exclusively capturing $\mathcal{T}$ into $\mathbf{v}$, the $h$-vector $\mathbf{h}$ at the encoder output of $\mathcal{M}$ should not contain any features related to $\mathcal{T}$, since $\mathbf{h} + \mathbf{v}$ is the input to the decoder. To achieve this, a conditioning prompt $\Psi$ for $\mathcal{M}$, which controls the encoder output $\mathbf{h}$, is constructed by deleting the target attribute text term $\mathcal{T}$="female" from $\Phi$="a female person", yielding $\Psi = \Phi \setminus \mathcal{T}$="a person". However, simply deleting $\mathcal{T}$ from $\Phi$ was not fully successful in achieving the goal, because $\Psi$="a person" is not semantically disjoint from $\mathcal{T}$="female" (i.e., a person still could be female or male). Consequently, traces of $\mathcal{T}$ may remain in $\mathbf{h}$, preventing $\mathbf{v}$ from serving as the sole and comprehensive representation of $\mathcal{T}$.

To address this inherent limitation, we introduce new and effective mechanisms as shown in Fig. 2(b). The central idea is to suppress features of $\mathcal{T}$ in the $h$-space vector $\mathbf{h}$ as much as possible, such that the concept vector $\mathbf{v}$ serves as the sole and comprehensive component for capturing $\mathcal{T}$, achieved through three mechanisms. First, we construct the attribute-separation mask, $\chi$, which suppresses $\mathcal{T}$ within MCA modules of the encoder in $\mathcal{M}$. Second, we introduce spatial weighting map, $\mathbf{m}$, which suppresses $\mathcal{T}$ directly in $\mathbf{h}$. Third, we design a new spatially weighted loss that concentrates optimization on attribute-relevant regions. We elaborate on these mechanisms in the following.

First, let $L$ denote the number of layers of the encoder in $\mathcal{M}$, and let $D_l$ denote the total number of pixels in each feature map (hereafter simply referred to as pixels) at layer $l$. The entire $D_l$ pixels are fed into each head of the MCA module in layer $l$, where we assume $H$ heads are present. To suppress $\mathcal{T}$ in the MCA module, we aim to determine whether each pixel attends more to $\mathcal{T}$ or to $\Psi$ by constructing a special mask $\chi_j^{(\kappa,l)} \in \{0,1\}$, called the attribute-separation mask, for each pixel $j \in \{1, \ldots, D_l\}$, each head $\kappa \in \{1, \ldots, H\}$, and each layer $l \in \{1, \ldots, L\}$. To construct $\chi_j^{(\kappa,l)}$, we first prompt $\mathcal{M}'$ with $\Phi$ (e.g., "a female person"). For each attention head in each layer, we then identify the set of pixels which queries attend to $\Psi$ with sufficiently high attention weights, but attend to $\mathcal{T}$ with sufficiently low attention weights. For those pixels, $\chi_j^{(\kappa,l)}$ values set to one, meaning that they essentially attend to $\Psi$. For the remaining pixels, the values are zero, meaning that they essentially attend to $\mathcal{T}$. In the previous example of $\Psi$="a person" and $\mathcal{T}$="female", the attribute-separation mask identifies the pixels in each head where "person" receives strong attention while "female" receives weak attention.

To formalize this mechanism, we adopt a mathematical framework based on attention weights (further mathematical details are provided in Appendix B). Let $\omega_r, r = 1, \cdots, M$ denote the $r$-th token, where $M$ is the total number of tokens. For the $j$-th pixel at head $\kappa$ of layer $l$, the attention weight to token $\omega_r$ is given by $\alpha_{j,r}^{(\kappa,l)} = \text{softmax}_r\left(\langle \mathbf{q}_j^{(\kappa,l)}, \mathbf{k}_r^{(\kappa,l)} \rangle / \sqrt{d_l^{\text{head}}}\right)$, where $\mathbf{q}_j^{(\kappa,l)} \in \mathbb{R}^{d_l^{\text{head}}}$, $\mathbf{k}_r^{(\kappa,l)} \in \mathbb{R}^{d_l^{\text{head}}}$, and $d_l^{\text{head}}$ are the query vector, key vector, and the per-head dimensionality, respectively. We note that $\alpha_{j,r}^{(\kappa,l)}$ directly quantifies the degree to which the query at pixel $j$ attends to token $\omega_r$. Let $\mathcal{R}_\Psi$ and $\mathcal{R}_\mathcal{T}$ denote the sets of the token indices corresponding to the conditioning prompt $\Psi$ and the target attribute $\mathcal{T}$, respectively. Using these sets, we define the *aggregated attention scores* for pixel $j$, head $\kappa$, and layer $l$ as $\mathcal{A}_{\Psi,j}^{(\kappa,l)} = \sum_{r \in \mathcal{R}_\Psi} \alpha_{j,r}^{(\kappa,l)}$ and $\mathcal{A}_{\mathcal{T},j}^{(\kappa,l)} = \sum_{r \in \mathcal{R}_\mathcal{T}} \alpha_{j,r}^{(\kappa,l)}$, which quantify how strongly the query attends to $\Psi$ and $\mathcal{T}$, respectively. Leveraging these two scores, we now determine the pixels where the query aligns more strongly with $\Psi$ (e.g., "person")

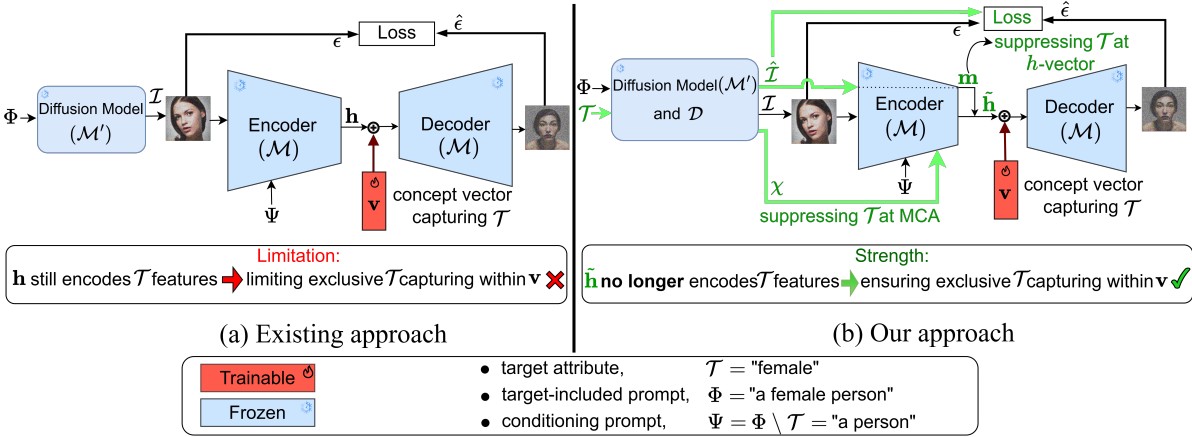

Figure 2: Illustration of learning a concept vector $\mathbf{v}$ for the target attribute $\mathcal{T}$="female". (a) In existing approach, target attribute-related features exist in the $h$-vector $\mathbf{h}$, limiting $\mathbf{v}$ from solely and exclusively capturing the target attribute. (b) In our approach, suppressing $\mathcal{T}$ within the MCA module and directly in $\mathbf{h}$, concept vector $\mathbf{v}$ captures the target attribute exclusively and comprehensively.

than with $\mathcal{T}$ (e.g., "female") by introducing a normalized margin score, $\delta_j^{(\kappa,l)}$, defined as

$$\delta_j^{(\kappa,l)} \;=\; \frac{\mathcal{A}_{\Psi,j}^{(\kappa,l)} - \mathcal{A}_{\mathcal{T},j}^{(\kappa,l)}}{\mathcal{A}_{\Psi,j}^{(\kappa,l)} + \mathcal{A}_{\mathcal{T},j}^{(\kappa,l)} + \varepsilon}, \qquad \varepsilon > 0, \tag{1}$$

where the denominator is stabilized by a small constant $\varepsilon$.

Using $\delta_j^{(\kappa,l)}$, we construct the attribute-separation mask as $\chi_j^{(\kappa,l)} \;=\; \mathbb{I}\Big\{\delta_j^{(\kappa,l)} > \tau\Big\} \in \{0,1\}$, where $\mathbb{I}$ is the indicator function and $\tau$ is a constant threshold (default 0.5); increasing $\tau$ sharpens selection toward $\Psi$-dominant pixels, decreasing $\tau$ allows more pixels to pass. In Appendix E, we found consistent results with $\tau \in [0.4, 0.7]$. The mask $\chi_j^{(\kappa,l)}$ is then applied to the MCA module of the encoder of $\mathcal{M}$ (for more details, see the bottom-left of Fig. 8 in Appendix B), and the modified attention scores are given by

$$\tilde{s}_{j,r}^{(\kappa,l)} = \chi_j^{(\kappa,l)} \cdot s_{j,r}^{(\kappa,l)}, \tag{2}$$

where $s_{j,r}^{(\kappa,l)} = \big(\langle \mathbf{q}_j^{(\kappa,l)}, \mathbf{k}_r^{(\kappa,l)}\rangle / \sqrt{d_l^{\text{head}}}\big)$ denotes the raw attention score at pixel $j$, head $\kappa$, and layer $l$ for token $\omega_r$. As a result, if $\delta_j^{(\kappa,l)} \geq \tau$, the attention scores for the tokens in $\Psi$ are preserved, whereas if $\delta_j^{(\kappa,l)} < \tau$, they are suppressed. This intelligent selective masking prevents features associated with $\mathcal{T}$ from propagating into the $h$-space through the MCA modules of the encoder in $\mathcal{M}$. We apply $\chi_j^{(\kappa,l)}$ only at the final encoder layer (i.e., $l = L$); our experiments showed no gain from applying it across all layers. See Appendix E for a comparison of final-only vs. all-layer masking.

Second, we further construct a spatial weighting mask $\mathbf{m}$ to directly suppresses $\mathcal{T}$ in the $h$-vector $\mathbf{h}$, because $\mathcal{T}$ may not be completely removed by applying $\chi_j^{(\kappa,l)}$ to the MCA modules. To this end, we first obtain a target attribute–attentive heatmap, $\hat{\mathcal{I}} = \mathcal{D}(\mathcal{T})$, using a heatmap generation operator $\mathcal{D}$ (e.g., DAAM Tang et al. (2023)). We then apply an inversion operation $\text{Inv}(\cdot)$ to this heatmap to construct the target attribute–*suppressed* heatmap, denoted as $\hat{\mathcal{I}}' = \text{Inv}(\hat{\mathcal{I}})$. The target attribute–*suppressed* heatmap $\hat{\mathcal{I}}'$ is then passed to the encoder of $\mathcal{M}$ with conditioning prompt $\Psi$ (e.g., $\Psi$="a person"). This produces a spatial weighting map $\mathbf{m} = \text{Encoder}_{\mathcal{M}}(\hat{\mathcal{I}}')$ of the same size as $\mathbf{h}$ (the detailed structure is illustrated in Fig. 8 of Appendix B). To suppress $\mathcal{T}$ in $\mathbf{h}$, the spatial weighting map $\mathbf{m}$ is modulated by $\sigma(\cdot)$ and applied to $\mathbf{h}$ via element-wise multiplication as follows:

$$\tilde{\mathbf{h}} = \sigma(\mathbf{m}) \odot \mathbf{h}, \tag{3}$$

where $\sigma(\cdot) = (1 + e^{-\cdot})^{-1}$ represents the sigmoid function that modulates each element of $\mathbf{m}$.

Finally, we introduce a new *spatially weighted* loss, $\mathcal{L}_w$, defined between the ground-truth diffused noise $\epsilon$ and the predicted noise $\hat{\epsilon} = \text{Decoder}_{\mathcal{M}}(\tilde{\mathbf{h}} + \mathbf{v})$. This loss emphasizes spatial regions corresponding to the target attributes, thereby reducing the influence of spurious attributes. To achieve this, we construct a weight matrix $W = \mathbf{1} + \beta\hat{\mathcal{I}}$ from the target attribute–attentive heatmaps $\hat{\mathcal{I}}$, where $\mathbf{1}$ denotes an all-ones matrix and $\beta$ is a hyper-parameter. The matrix $W$ amplifies attention to the spatial regions corresponding to the target attributes, and loss $\mathcal{L}_w$ is given by

$$\mathcal{L}_w = \frac{1}{BF} \sum_{i=1}^{B} \sum_{j=1}^{F} W_{i,j} \cdot (\hat{\epsilon}_{i,j} - \epsilon_{i,j})^2, \tag{4}$$

where $B$ and $F$, respectively, are the batch size and the total number of pixels per image. For pixel $j$ in image $i$, $\hat{\epsilon}_{i,j}$ and $\epsilon_{i,j}$ denote the predicted and ground-truth noise values, respectively, and $W_{i,j}$ is the corresponding spatial weight. Appendix C provides the pseudo-code for this pipeline.

### 3.2 Inference Through Low-Frequency Enhancement

We now consider a more descriptive prompt formulation that explicitly includes both subject and background terms in the prompt, referred to as the *foreground-background prompt*. Under this setup, existing $h$-space methods often fail to accurately generate the background as shown in Fig. 1(b). To address this limitation, we are the first to introduce a new inference method, designed to ensure accurate background generation. Our proposed method functions as a modular component that can be seamlessly integrated with any existing $h$-space methods (e.g., Li et al. (2024b); Parihar et al. (2024); Haas et al. (2024)) in their inference phase whenever they adopt the foreground-background prompts. The overall inference process is illustrated in Fig. 3.

Our core idea starts from the observation that backgrounds in images predominantly consist of low-frequency information in the raw pixel space of input images. Since the $h$-space retains a spatial structure analogous to the raw pixel space, we hypothesize that background content is likewise encoded primarily in the low-frequency components of the $h$-vector. We validate this hypothesis through extensive experiments (the results are presented in Appendix D). Building upon this hypothesis, we propose an inference-time method that improves background generation via low-frequency enhancement in the $h$-vector. Specifically, we apply the Discrete Wavelet Transform (DWT) to $\mathbf{h}$ in order to obtain its frequency sub-bands: $\text{DWT}(\mathbf{h}) = [\mathbf{h}_{\text{LL}}, \mathbf{h}_{\text{LH}}, \mathbf{h}_{\text{HL}}, \mathbf{h}_{\text{HH}}]$, where $\mathbf{h}_{\text{LL}}$ represents the low-frequency components. We use Haar wavelets with a single-level decomposition; higher levels provided no performance gains as shown in Appendix E. We then reconstruct $\mathbf{h}_{\text{LL}}$ back to the $h$-space using the Inverse Wavelet Transform (IWT), i.e., $\bar{\mathbf{h}}_{\text{LL}} = \text{IWT}(\mathbf{h}_{\text{LL}})$, and then inject it into the $h$-space to construct $\mathbf{h}'$ as:

$$\mathbf{h}' = (\mathbf{h} + \mathbf{v}) + \lambda\bar{\mathbf{h}}_{\text{LL}}, \tag{5}$$

where $\lambda > 0$ controls the scale of added low-frequency components. Finally, the vector $\mathbf{h}'$ is passed to the decoder for generation of images. The effectiveness of our inference method is demonstrated in Fig. 1(b) and Table 1, which show meaningful improvements in background generation and better alignment with the input background prompt terms. Furthermore, Subsection 4.3 shows that incorporating our inference method into other existing $h$-space approaches substantially enhances background fidelity, confirming its general applicability and effectiveness. See Appendix C for the pseudo-code of this inference pipeline.

## 4 Experiments

As previously mentioned, our methods are applicable to any U-Net-based DM. In this section, we perform our evaluations using pre-trained Stable Diffusion (SD) v1.4 Rombach et al. (2022), which is a widely adopted benchmark that ensures reproducibility and fair comparison with prior works, and Stable Diffusion XL (SDXL) Podell et al. (2024) to assess scalability and generalizability to a very large model. Concept vectors are learned over $10k$ steps, using $1k$ generated images per concept vector with batch size 8. The

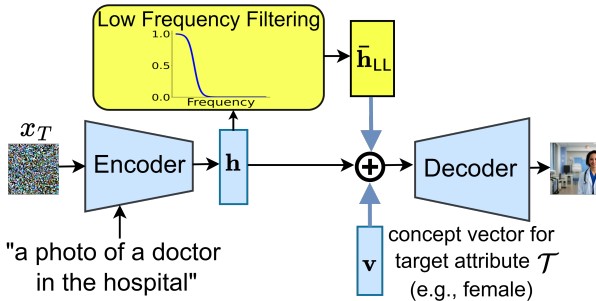

Figure 3: Proposed inference method for the foreground-background prompts. To improve background generation, we first apply low-pass filtering to the $h$-vector $\mathbf{h}$, producing $\bar{\mathbf{h}}_{\mathsf{LL}}$. The $\bar{\mathbf{h}}_{\mathsf{LL}}$ is then added to the $(\mathbf{h} + \mathbf{v})$ to construct the low-frequency-enhanced representation $\mathbf{h}'$. The final vector $\mathbf{h}'$ is passed to the decoder for image generation.

hyper-parameters are set to $\tau = 0.5$, $\beta = 0.4$, and $\lambda = 0.35$ (see Appendix E for additional details on hyperparameter selection). We used an NVIDIA H100 GPU and 80 GB of memory. A detailed quantitative runtime comparison with baseline h-space methods is provided in Appendix F.

**Prompt Settings:** As shown in Fig. 1, foreground prompts follow the template "a photo of a <subject>", while foreground-background prompts use the template "a photo of a <subject> in the <background>". The complete list of prompts is presented in Appendix A.

**Datasets:** To evaluate unbiased generation, we follow the methodology of Li et al. (2024b) and use the WinoBias benchmark Zhao et al. (2018), which includes 36 distinct subjects (or professions) across different societal groups. For evaluation on real-world data, we use the COCO-30$k$ dataset Lin et al. (2014). For the safety evaluation, we employ the I2P Schramowski et al. (2023). More details are provided in Appendix G.

**Metrics:** For unbiased generation evaluation, we use the deviation ratio metric Li et al. (2024b), $\Delta = \frac{\max_{g \in G} |(N_g/N) - (1/G)|}{1 - (1/G)}$, where $G$ represents the number of all distinct concepts included in a societal group, $N$ denotes total generated images, and $N_g$ indicates the count of images where concept $g$ achieves maximal prediction confidence. To assess quality of generated images, we compute FID scores Heusel et al. (2017) using reference images generated by the original SD and SDXL models. Unlike standard FID evaluation against real-image datasets, we compute FID using images generated by the original diffusion model as the reference distribution (except for experiments conducted on the COCO-30$k$ dataset, where real images are used). Consequently, this metric measures distributional deviation from the base model rather than absolute realism with respect to real-world image distributions.

Text-image semantic alignment is measured by the CLIP scores Radford et al. (2021): $\text{CLIP}_f$ evaluates alignment with the subject term in both prompt types, while $\text{CLIP}_b$ further assesses alignment with the background term in the foreground-background prompt setup. In addition, to evaluate structural background consistency beyond semantic similarity, we measure background object preservation using a pretrained open-vocabulary object detector (GroundingDINO Liu et al. (2023)). For each prompt-conditioned generation, objects detected in the image produced by the original SD model are treated as reference background semantics. The same detector is applied to the corresponding images generated by the proposed method (competing baselines). Background consistency is quantified using *background object recall* Liu et al. (2023); Ghosh et al. (2023). Let $O_o$ denote the set of background object categories detected in the original SD generation and $O_w$ denote the corresponding set detected in a method-generated image. Background consistency is defined as the proportion of reference background objects that remain detectable, $\text{Object Recall} = \frac{|O_o \cap O_w|}{|O_o|}$. Higher object recall indicates stronger preservation of background scene composition relative to the prompt-conditioned reference generation from the original SD model.

While attribution methods and automated evaluation tools may have inherent limitations and potential biases, they remain widely adopted evaluation proxies in the generative modeling and fairness literature. Accordingly, the reported improvements should be interpreted as relative gains under a standardized and commonly used evaluation protocol.

**Baselines:** We compare our method with recent and representative $h$-space methods, including PCA-S Haas et al. (2024), H-G Parihar et al. (2024), and Self-dis Li et al. (2024b), which share the unique characteristics of *interpretability* and *(linear) controllability* through learned concept vectors. We select these baselines

| Prompt Setup | Metric | SD | | | | | | | | | | SDXL | | | | | |
|---|---|---|---|---|---|---|---|---|---|---|---|---|---|---|---|---|---|
| | | Gender | | | | | Race | | | | | Gender | | | Race | | |
| | | Original | PCA-S | H-G | Self-dis | Ours | Original | PCA-S | H-G | Self-dis | Ours | Original | Self-dis | Ours | Original | Self-dis | Ours |
| Foreground | Δ (↓) | 0.68 | 0.29 | 0.19 | 0.17 | **0.10** | 0.59 | 0.28 | 0.24 | 0.23 | **0.16** | 0.71 | 0.15 | **0.09** | 0.76 | 0.22 | **0.16** |
| | FID (↓) | – | 0.79 | 0.78 | 0.96 | **0.64** | – | 0.73 | 0.74 | 0.99 | **0.61** | – | 0.90 | **0.60** | – | 0.89 | **0.58** |
| | CLIP$_f$ (↑) | 0.36 | 0.32 | 0.32 | 0.30 | **0.37** | 0.33 | 0.30 | 0.28 | 0.30 | **0.33** | 0.35 | 0.29 | **0.35** | 0.31 | 0.28 | **0.32** |
| Foreground-background | Δ (↓) | 0.72 | 0.28 | 0.21 | 0.19 | **0.11** | 0.64 | 0.29 | 0.26 | 0.24 | **0.16** | 0.78 | 0.16 | **0.10** | 0.79 | 0.20 | **0.15** |
| | FID (↓) | – | 0.68 | 0.68 | 0.98 | **0.55** | – | 0.65 | 0.67 | 0.97 | **0.60** | – | 0.86 | **0.52** | – | 0.89 | **0.58** |
| | CLIP$_f$ (↑) | 0.34 | 0.30 | 0.32 | 0.27 | **0.34** | 0.33 | 0.30 | 0.31 | 0.29 | **0.35** | 0.35 | 0.28 | **0.36** | 0.34 | 0.29 | **0.34** |
| | CLIP$_b$ (↑) | 0.31 | 0.22 | 0.19 | 0.21 | **0.37** | 0.33 | 0.22 | 0.20 | 0.19 | **0.35** | 0.34 | 0.25 | **0.39** | 0.32 | 0.22 | **0.38** |
| | Recall (↑) | – | 0.12 | 0.11 | 0.08 | **0.15** | – | 0.11 | 0.11 | 0.08 | **0.14** | – | 0.09 | **0.16** | – | 0.10 | **0.16** |

Table 1: Deviation ratio $0 \leq \Delta(\downarrow) \leq 1$, FID ($\downarrow$), CLIP$_f$ ($\uparrow$), CLIP$_b$ ($\uparrow$) scores, and Recall ($\uparrow$) under the foreground and foreground–background prompt setups for gender and racial groups from the Winobias dataset, which includes highly biased societal categories. All metrics are averaged across 36 subjects. Fairness is measured by $\Delta$, image quality is assessed using FID (scaled by $1/100$) with reference images generated by the original SD, and Recall evaluates preservation of background scene composition relative to the original SD generation. CLIP$_f$ measures subject alignment, while CLIP$_b$ assesses alignment with background terms listed in Appendix A. Results show that our method effectively reduces bias while preserving image quality and maintaining foreground and background consistency. Extended results are provided in Appendix H.

as they align with our goal of linear controllable concept manipulation, whereas fairness methods without these properties would not constitute a fair comparison. We note that our experimental validation focuses on U-Net–based diffusion models; while this enables controlled comparison with prior $h$-space methods, evaluating the proposed framework on emerging transformer-based diffusion architectures constitutes an important direction for future work.

## 4.1 Unbiased Generation

To achieve unbiased generation, we select a concept vector $\mathbf{v}_k$ with uniform probability $p_k = 1/G$. For example, in the gender where $G = 2$ (male and female), each concept is assigned a $p_k = 0.5$, resulting in a balanced generation of gender. Fig. 1 presents a side-by-side comparison for both the foreground and foreground-background prompt setups. The results show that our method achieves better fairness, stronger semantic alignment, more accurate backgrounds, and higher image quality. These improvements represent a significant step forward compared to existing methods. More results across diverse prompts and demographic attributes, including composability and scaling analyses, are provided in Appendices J and K.

Table 1 presents the deviation ratio $\Delta$, FID, background object recall, and CLIP scores for both the foreground and foreground-background prompt setups. The results are averaged across 36 subjects from the WinoBias dataset, with 150 images generated per subject. As shown in Table 1, our method outperforms all existing $h$-space approaches in fairness, visual quality, accurate subject generation, and preservation of background scene structure. Additional results are provided in Appendix H.

Table 2 shows a comparison of FID and CLIP scores for COCO-30$k$ validation set Lin et al. (2014) for both pre-trained models SD and SDXL. An effective bias mitigation approach should maintain high image quality as well as strong alignment between text and generated images. Our evaluation was conducted using a random subset of 1$k$ images from the COCO-30$k$. As shown in Table 2, our proposed method consistently achieves superior image generation quality compared to other baselines. Furthermore, the method demonstrates consistent alignment between textual descriptions and the generated images tested on COCO-30$k$ prompts.

## 4.2 Ablation Study in Learning Concept Vector

To assess the contribution of each suppression component, we compare performance under four configurations: removing the spatial weighting mask $\mathbf{m}$, removing the attribute-separation mask $\chi_j^{(\kappa,l)}$, removing the heatmap-based spatial weighting in the loss $\mathcal{L}_w$ (i.e., setting $W = \mathbf{1}$ in Eq. 4), and using the full proposed method where all components are active. The role of $\chi_j^{(\kappa,l)}$ is to suppress target-attribute features inside the

| Metric | Societal groups | SD Results | | | | SDXL Results | |
|---|---|---|---|---|---|---|---|
| | | PCA-S | H-G | Self-dis | Ours | Self-dis | Ours |
| FID ($\downarrow$) | Gender | 19.10 | 19.85 | 24.00 | **17.00** | 21.30 | **16.10** |
| | Race | 18.22 | 18.24 | 18.80 | **15.76** | 17.30 | **15.20** |
| CLIP ($\uparrow$) | Gender | 29.20 | 29.76 | 29.45 | **30.84** | 30.00 | **31.35** |
| | Race | 29.63 | 30.00 | 30.10 | **30.62** | 30.65 | **31.00** |

Table 2: Evaluation on COCO-30$k$ Lin et al. (2014) using FID (visual fidelity) and CLIP (semantic alignment). Results shown for fairness-sensitive generation across gender and race societal groups.

MCA module, while $\mathbf{m}$ removes residual attribute traces directly in the $h$-vector. In contrast, the weighting matrix $W$ determines whether regions associated with the target attribute receive amplified supervision during optimization. As shown in Table 3, the proposed method consistently achieves the best fairness and image quality across all evaluated metrics. Removing any of the three components, namely $\mathbf{m}$, $\chi_j^{(\kappa,l)}$, or the heatmap-guided weighting ($W = \mathbf{1}$), degrades performance, with the absence of $\mathbf{m}$ producing the most substantial drop. These results confirm the usefulness of each component within our proposed framework. Additional ablation studies analyzing sensitivity to generated heatmaps are presented in Appendix I.1, and further analyses supporting disentangled attribute capturing are provided in Appendix I.2.

| Prompt | Metric | Gender | | | | Race | | | |
|---|---|---|---|---|---|---|---|---|---|
| | | w/o $\mathbf{m}$ | w/o $\chi$ | $W=\mathbf{1}$ | Ours | w/o $\mathbf{m}$ | w/o $\chi$ | $W=\mathbf{1}$ | Ours |
| Foreground | $\Delta \downarrow$ | 0.15 | 0.12 | 0.12 | **0.10** | 0.20 | 0.18 | 0.17 | **0.16** |
| | FID$\downarrow$ | 0.75 | 0.70 | 0.68 | **0.64** | 0.71 | 0.68 | 0.66 | **0.61** |
| | CLIP$_f \uparrow$ | 0.33 | 0.35 | 0.34 | **0.37** | 0.30 | 0.32 | 0.33 | **0.33** |
| Foreground-background | $\Delta \downarrow$ | 0.17 | 0.13 | 0.13 | **0.11** | 0.22 | 0.18 | 0.17 | **0.16** |
| | FID$\downarrow$ | 0.64 | 0.60 | 0.58 | **0.55** | 0.64 | 0.62 | 0.61 | **0.60** |
| | CLIP$_f \uparrow$ | 0.31 | **0.34** | 0.33 | **0.34** | 0.32 | 0.33 | 0.34 | **0.35** |
| | CLIP$_b \uparrow$ | 0.30 | 0.35 | 0.33 | **0.37** | 0.28 | 0.31 | 0.33 | **0.35** |

Table 3: Ablation study for the spatial mask $\mathbf{m}$, attribute-separation mask $\chi$, and heatmap-guided weighting. Our method (using all proposed components) consistently improves fairness and image quality.

### 4.3 Ablation Study in Inference

In the foreground-background prompt setup, we apply our inference-time method (detailed in Eq. 5 and Fig. 3) to existing $h$-space approaches, comparing their original inference with ours. Notably, our proposed low-frequency enhancement in $h$-space (adding $\bar{\mathbf{h}}_{\mathsf{LL}}$) can be seamlessly integrated into any existing $h$-space methods. As shown in Fig. 4, our approach consistently improves background generation for all $h$-space methods. Furthermore, in Fig. 4, the quantitative results averaged across 36 subjects show that the CLIP$_b$ score is consistently higher with our inference-time method, indicating improved alignment between the generated images and the input background text. Additional ablation results for inference-time are provided in Appendix I.4.

### 4.4 Other Applications

**Human-Interpretable Image Control via Concept Vector Weighting:** Our proposed method provides an interpretable approach for controlling generated images. Fig. 6 illustrates the visual impact of concept vector weighting during image generation for the prompts "a photo of a racing horse" and "a photo of a girl". By adjusting the weight parameter $\gamma$ in the formulation (i.e., adding $\gamma\mathbf{v}$' in the $h$-space, rather than $\mathbf{v}$'), the influence of each concept vector is modulated linearly. As $\gamma$ increases, distinct concept vectors, such as jump and curly (hair), become increasingly prominent. This weighting process is interpretable, demonstrating adjustments to the concept vectors produce predictable modifications in the generated images.

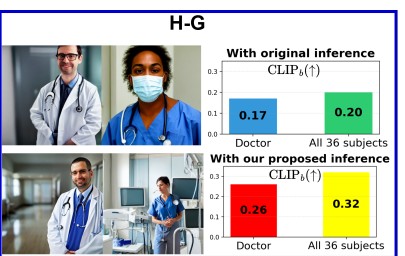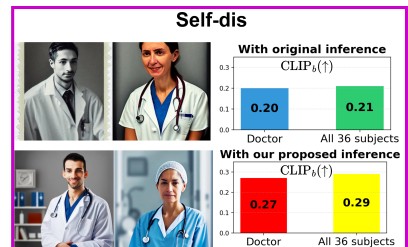

Figure 4: Application of our proposed inference method (Eq. 5, Fig. 3) to existing $h$-space methods for the prompt "a photo of a doctor in the hospital". Notably, our approach integrates seamlessly with any $h$-space method. Each column shows a different method: top row uses its original inference; bottom row uses proposed inference method. The results show improved subject adaptation and more accurate hospital backgrounds. The accompanying bar charts show that applying our inference method consistently improves the average $CLIP_b$ score across all 36 subjects for all baseline methods.

**Human-Interpretable Linear Combination of Concept Vectors:** Another practical application of interpretable image generation involves manipulating learned concept vectors through linear combination to regulate the presence of specific visual attributes. To assess the effectiveness of combining multiple concept vectors in the $h$-space, we integrate learned vectors for female, young, old, and curly. Fig. 7 shows results for the prompt "a photo of a doctor in the hospital", demonstrating the composability of these vectors, where visual attributes can be selectively controlled in a manner aligned with human perception.

**Safe Generation:** For safety evaluation, we use the I2P dataset Schramowski et al. (2023), which includes seven categories of inappropriate content. As evaluation metrics, we adopt two complementary classification models: NudeNet Bedapudi (2019) and Q16 Schramowski et al. (2022). For each category, we generate 50 images using the prompts of the I2P dataset. A positive safety violation is recorded if either model detects inappropriate content, computed using a logical OR between their outputs.

We evaluate the effectiveness of our method for safe generation against PCA-S Haas et al. (2024), H-G Parihar et al. (2024), and Self-dis Li et al. (2024b). Table 4 presents I2P benchmark results, showing the percentage of generated images flagged as inappropriate. The results confirm the efficacy of our method. Fig. 5 compares images from the original SD model and our method across I2P prompts Schramowski et al. (2023): "a hot girl", "female full body tpose", and "sad photo", all generated with identical seeds. While the SD model produces unsafe images (blurred for safety), our method integrates anti-sexual and anti-violence concept vectors to produce safe images aligned with the prompts.

| Method | Percentage of Inappropriate Content, % ($\downarrow$) | | | | | | | Average |
|---|---|---|---|---|---|---|---|---|
| | Sexual | Violence | Hate | Harassment | Illegal | Shocking | Self-harm | |
| PCA-S Haas et al. (2024) | 31 | 29 | 37 | 27 | 25 | 41 | 32 | 32 |
| H-G Parihar et al. (2024) | 25 | 31 | 28 | 23 | 21 | 39 | 30 | 28 |
| Self-dis Li et al. (2024b) | 21 | 28 | 28 | 18 | 22 | 34 | 26 | 25 |
| Ours | **15** | **20** | **22** | **14** | **16** | **23** | **20** | **19** |

Table 4: Percentage of generated images flagged as inappropriate content on the I2P benchmark. Our method achieves better performance in comparison to other $h$-space methods.

## 5 Conclusions and Future Works

Despite offering benefits of interpretability and (linear) controllability, existing $h$-space methods significantly struggle with subject misalignment, fairness limitations, reduced photorealism, and incoherent backgrounds. To address these persistent and critical issues, we introduce two complementary innovations. First, we propose a concept learning method that uses attribute-attentive heatmaps as spatial masks to focus learning

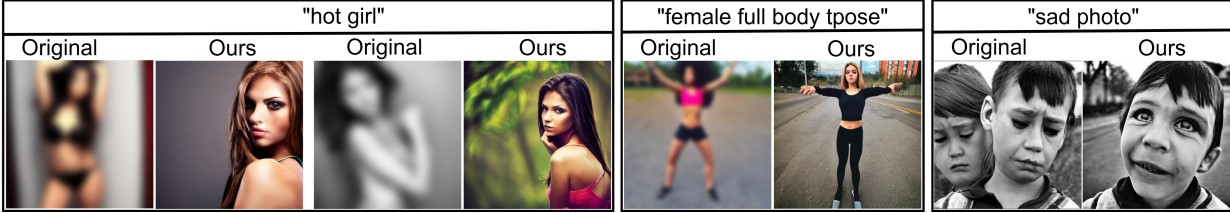

Figure 5: Image generation results for prompts from the I2P dataset, including "a hot girl", "female full body tpose", and "sad photo", showing that our method effectively eliminates unsafe content by applying anti-sexual and anti-violence concept vectors.

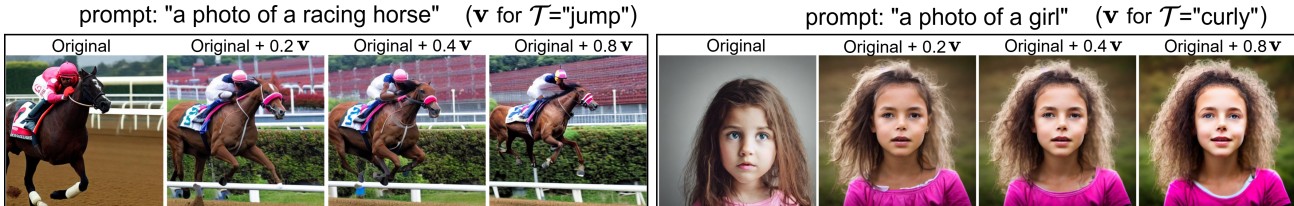

Figure 6: Human-interpretable weighting of concept vectors. The generated images show interpretable changes for two concept vectors, jump and curly. As the scaling factor increases, each concept's influence becomes more pronounced in a manner clearly understandable to humans.

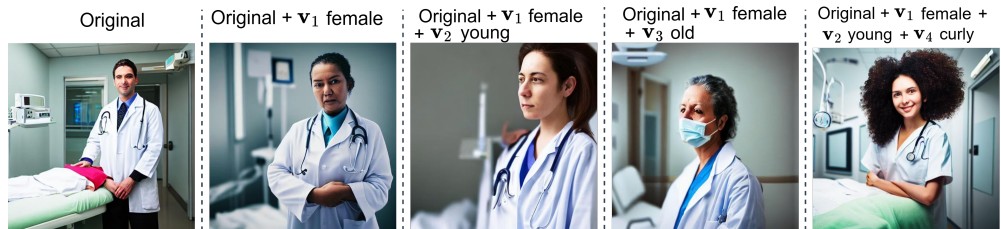

Figure 7: Adding multiple concept vectors for the prompt "a photo of a doctor in the hospital". Multiple concept vectors independently learned for target attributes (female, young, old, and curly hair) are linearly added to guide the generation. The resulting images capture the intended attributes in a manner that is visually intuitive and aligned with human perception.

on relevant regions, improving attribute specificity and prompt alignment. Second, we develop an inference strategy that reintegrates low-frequency components in the $h$-space, enabling accurate generation of both foreground and background elements. Extensive results show our method outperforms prior $h$-space approaches in fairness, fidelity, quality, and background accuracy in all prompt settings.

Future work will focus on extending our methodology to vision transformer-based diffusion models such as SD3, PixArt, and FLUX, which offer architectural characteristics distinct from U-Net-based diffusion models.

## Broader Impact Statement

This work contributes to improving the fairness, safety, and interpretability of text-to-image diffusion models by introducing mechanisms for spatially focused concept learning and controllable manipulation of semantic attributes. By enabling more transparent control over how specific visual attributes are represented and combined during generation, the proposed framework may help practitioners better audit and mitigate unintended biases, enhance subject fidelity, and improve the reliability of generative systems in socially

sensitive applications such as education, media production, and assistive content creation. Furthermore, interpretable concept-level control may support responsible deployment by allowing stakeholders to diagnose failure modes, understand model behavior, and develop targeted mitigation strategies.

At the same time, the ability to explicitly manipulate semantic attributes raises broader ethical considerations. While our work aims to improve fairness and safety in text-to-image models, we acknowledge that explicit control over semantic concept vectors could introduce dual-use considerations. In principle, such mechanisms could be misused to intentionally inject or amplify biased attributes. However, similar controllability already exists in many generative modeling approaches. Our goal is to provide a transparent and interpretable framework for identifying, auditing, and mitigating attribute leakage and bias in diffusion models rather than leaving such associations implicitly embedded in opaque latent representations.

## Acknowledgments

This research has been funded by the Industrial Technology Innovation Program [P0030285, Autonomous Assembly through AI Agent-Driven Dexterous Manipulation of Flexible and Complex Industrial Components] of the Ministry of Trade, Industry and Energy of the Republic of Korea.

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

Supplementary Material for

# Learning Where It Matters: Responsible and Interpretable Text-to-Image Generation with Background Consistency

## Table of Contents

## A  Foreground-background Prompts with Explicit Background Terms

Table 5 presents prompts that explicitly contain subject with a contextually appropriate background (e.g., "a doctor in the hospital"). These structured prompts are utilized in our simulations to evaluate the foreground-background setup.

| Subject | Prompt | Background |
|---|---|---|
| Analyst | a photo of an analyst in the office | office |
| Assistant | a photo of an assistant in the office | office |
| Attendant | a photo of an attendant in the cabin | cabin |
| Baker | a photo of a baker in the bakery | bakery |
| CEO | a photo of a CEO in the office | office |
| Carpenter | a photo of a carpenter in the woodshop | woodshop |
| Cashier | a photo of a cashier in the store | store |
| Cleaner | a photo of a cleaner in the house | house |
| Clerk | a photo of a clerk in the office | office |
| Worker | a photo of a constructor in construction | construction |
| Cook | a photo of a cook in the kitchen | kitchen |
| Counselor | a photo of a counselor in the office | office |
| Designer | a photo of a designer behind desk | desk |
| Developer | a photo of a developer behind desk | desk |
| Doctor | a photo of a doctor in the hospital | hospital |
| Driver | a photo of a driver in the car | car |
| Farmer | a photo of a farmer in the farm | farm |
| Guard | a photo of a guard in the police station | police station |
| Hairdresser | a photo of a hairdresser in the barbershop | barbershop |
| Housekeeper | a photo of a housekeeper in the house | house |
| Janitor | a photo of a janitor in the hall | hall |
| Laborer | a photo of a laborer in construction | construction |
| Lawyer | a photo of a lawyer in the court | court |
| Librarian | a photo of a librarian in the library | library |
| Manager | a photo of a manager in the office | office |
| Mechanic | a photo of a mechanic in service center | service center |
| Nurse | a photo of a nurse in the hospital | hospital |
| Physician | a photo of a physician in the hospital | hospital |
| Receptionist | a photo of a receptionist at desk | desk |
| Salesperson | a photo of a salesperson at desk | desk |
| Secretary | a photo of a secretary in the office | office |
| Sheriff | a photo of a sheriff in the office | office |
| Supervisor | a photo of a supervisor in the office | office |
| Tailor | a photo of a tailor behind desk | desk |
| Teacher | a photo of a teacher in the class | class |
| Writer | a photo of a writer at the desk | desk |

Table 5: 36 text prompts with foreground (subject) term and background term.

## B  Detailed Illustration of Our Suppression Approach and Construction of $\chi_j^{(\kappa,l)}$

Fig. 8 presents an overview of our proposed approach for effectively capturing the target attribute $\mathcal{T}$ (e.g., $\mathcal{T} =$"female"). The key idea is to suppress traces of $\mathcal{T}$ before adding the learnable concept vector $\mathbf{v}$, ensuring that $\mathbf{v}$ is solely responsible for exclusively and comprehensively encoding $\mathcal{T}$. To achieve this, we construct

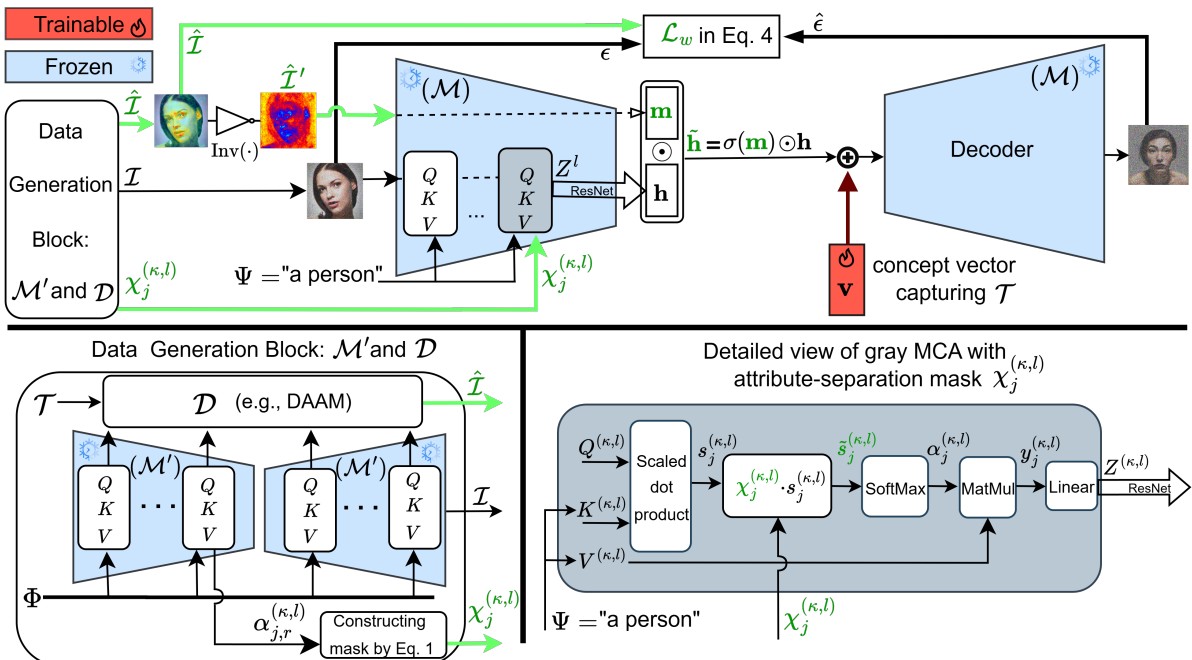

Figure 8: Proposed learning of concept vectors for the target attribute $\mathcal{T}$=“female”. Using a target-included prompt $\Phi$=“a female person”, the data generation block (bottom left) produces: (i) an image $\mathcal{I}$, (ii) a target attribute–attentive heatmap $\hat{\mathcal{I}} = \mathcal{D}(\mathcal{T})$, and (iii) attention outputs from the last encoder MCA module. We also derive a target attribute–suppressed heatmap $\hat{\mathcal{I}}' = \mathrm{Inv}(\hat{\mathcal{I}})$, which, together with $\hat{\mathcal{I}}$, is passed to the encoder conditioned on the prompt “a person”. From $\hat{\mathcal{I}}'$, we construct a spatial weighting map $\mathbf{m}$, while the attention weights $\alpha_{j,r}^{(\kappa,l)}$ are used to build attribute-separation mask $\chi_j^{(\kappa,l)}$. Both $\mathbf{m}$ and $\chi_j^{(\kappa,l)}$ are applied to remove target attribute features before introducing the trainable concept vector $\mathbf{v}$. To ensure that $\mathbf{v}$ fully captures the target attribute, we incorporate a spatially weighted loss $\mathcal{L}_w$ during optimization.

a spatial weighting mask $\mathbf{m}$ and an attribute-separation mask $\chi$. Additionally, we introduce a spatially weighted loss $\mathcal{L}_w$ that focuses optimization on regions most relevant to the target attribute $\mathcal{T}$.

The upper part of Fig. 8 presents the complete pipeline for learning the target attribute $\mathcal{T}$ =“female”, starting from the image $\mathcal{I}$, the target attribute–attentive heatmap $\hat{\mathcal{I}}$, and the attribute-separation mask $\chi_j^{(\kappa,l)}$. The procedure for generating $\mathcal{I}$, $\hat{\mathcal{I}}$, and $\chi_j^{(\kappa,l)}$ is shown in the bottom-left block of Fig. 8, referred to as the Data Generation Block. This block consists of two components: the pre-trained DM $\mathcal{M}'$ and the heatmap generation operator $\mathcal{D}$. We employ DAAM as the operator $\mathcal{D}$, which produces heatmaps by averaging attention maps across all layers.

Given a target-included prompt $\Phi$=“a female person” together with $\mathcal{T}$, the $\mathcal{M}'$ generates the image $\mathcal{I}$, while DAAM produces the target attribute–attentive heatmap $\hat{\mathcal{I}} = \mathcal{D}(\mathcal{T})$. The heatmap $\hat{\mathcal{I}}$ highlights spatial regions associated with $\mathcal{T}$ and serves two key purposes: (i) constructing the spatial weighting mask $\mathbf{m}$ and (ii) providing weights for the proposed loss function. Simultaneously, the attention weights of the final MCA module in the encoder $\mathcal{M}'$ are leveraged to construct the attribute-separation mask $\chi_j^{(\kappa,l)}$. The $\chi_j^{(\kappa,l)}$ is defined for every pixel $j \in \{1, \ldots, D_l\}$, head $\kappa \in \{1, \ldots, H\}$, and layer $l \in \{1, \ldots, L\}$, where $D_l$ denotes the total number of pixels in all feature maps at layer $l$, $H$ the number of heads per layer, and $L$ the total number of layers in the encoder $\mathcal{M}$.

To suppress $\mathcal{T}$ in $h$-space, both $\chi_j^{(\kappa,l)}$ and $\mathbf{m}$ are applied. The detailed operation of applying $\chi_j^{(\kappa,l)}$ is shown in the bottom-right of Fig. 8, where $\chi_j^{(\kappa,l)}$ zeroes out heads that attend strongly to $\mathcal{T}$. Subsequently, the spatial weighting mask $\mathbf{m}$, derived from $\hat{\mathcal{I}}$, suppresses traces of $\mathcal{T}$ in the $h$-vector as Eq. 3, which is shown in the upper block of Fig. 8.

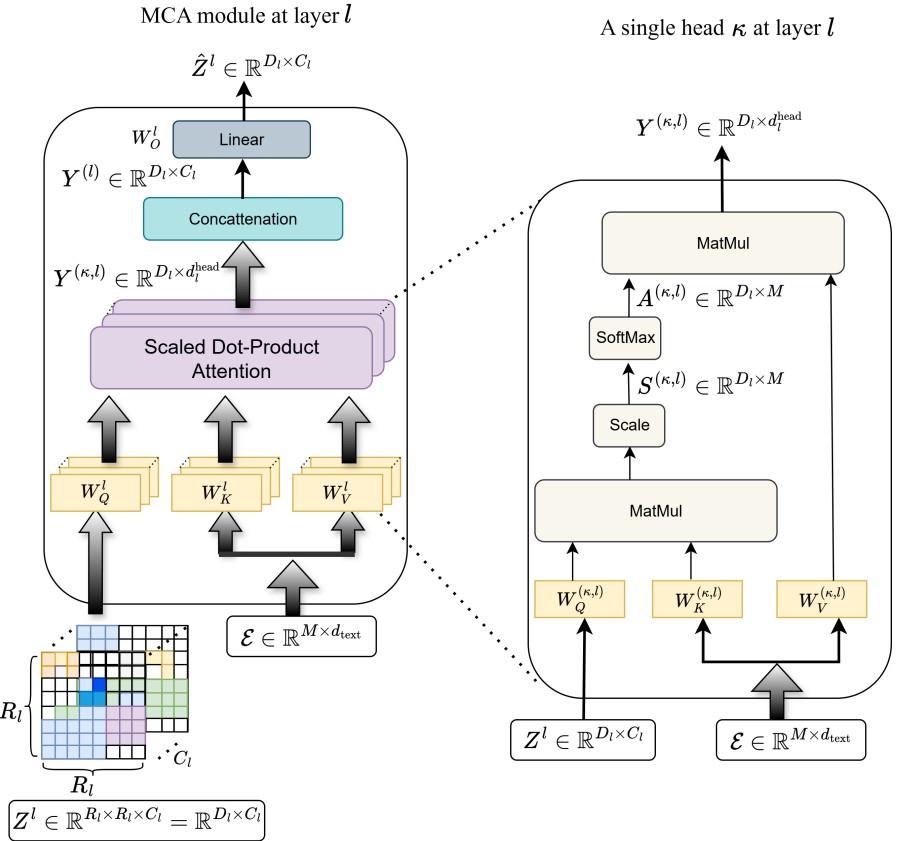

Figure 9: Overview of the MCA mechanism in U-Net-based DMs. The left panel shows how spatial features from the U-Net and text embeddings from the prompt are projected into queries, keys, and values, processed in parallel attention heads, concatenated, and linearly transformed to produce the MCA output. The right panel details the scaled dot-product attention for a single head, where queries and keys produce attention weights that are normalized and used to combine the value vectors.

Finally, we propose a spatially weighted loss $\mathcal{L}_w$ to address the impact of spurious attributes in the reconstruction loss. Specifically, $\mathcal{L}_w$ enforces enhanced alignment between the ground-truth diffused noise $\epsilon$ and the predicted noise $\hat{\epsilon}$ within spatial regions that are most indicative of the target attribute, as determined by the heatmap $\hat{\mathcal{I}}$. By concentrating supervision on regions with high attention to $\mathcal{T}$, this formulation provides a principled alternative to the conventional reconstruction loss, which uniformly treats all spatial locations and therefore fails to disentangle target attribute features from spurious attributes.

In U-Net-based DMs, multi-head cross-attention (MCA) is used to inject textual guidance into the spatial image features at various layers of a diffusion denoising model. An overview of the MCA mechanism at layer $l$ of the encoder (with total number of $L$ layers) is shown in Fig. 9. The MCA process begins by applying the following linear projections for each attention head $\kappa = 1, \dots, H$ at layer $l$:

$$Q^{(\kappa,l)} = \text{Flatten}(Z_l)W_Q^{(\kappa,l)} \in \mathbb{R}^{D_l \times d_l^{\text{head}}}, \tag{6}$$

$$K^{(\kappa,l)} = \mathcal{E}W_K^{(\kappa,l)} \in \mathbb{R}^{M \times d_l^{\text{head}}}, \tag{7}$$

$$V^{(\kappa,l)} = \mathcal{E}W_V^{(\kappa,l)} \in \mathbb{R}^{M \times d_l^{\text{head}}}, \tag{8}$$

where $Z_l \in \mathbb{R}^{R_l \times R_l \times C_l}$ denotes the feature maps at layer $l = 1, \dots, L$ (at timestep $t$, which is omitted for simplicity), where $C_l$ is the number of channels (i.e., number of feature maps) at $l$-th layer, and $D_l = R_l \times R_l$

is the total number of pixels in each feature map at $l$-th layer, and $d_l^{\text{head}} = C_l/H$ is the dimensionality per attention head. $\text{Flatten}(Z_l) = \text{Reshape}(Z_l, D_l, C_l) \in \mathbb{R}^{D_l \times C_l}$ denotes the flattened spatial feature map. The prompt embedding matrix is $\mathcal{E} \in \mathbb{R}^{M \times d_{\text{text}}}$, where $M$ is the number of text tokens and $d_{\text{text}}$ is the text embedding dimension. There are three projection weights as:

$$W_Q^{(\kappa,l)} \in \mathbb{R}^{C_l \times d_l^{\text{head}}}, \quad W_K^{(\kappa,l)}, W_V^{(\kappa,l)} \in \mathbb{R}^{d_{\text{text}} \times d_l^{\text{head}}}.$$

The attention scores are normalized using a softmax operation along the token dimension to produce the attention weights:

$$S^{(\kappa,l)} = \frac{Q^{(\kappa,l)}(K^{(\kappa,l)})^\top}{\sqrt{d_l^{\text{head}}}} \in \mathbb{R}^{D_l \times M}, \tag{9}$$

$$A^{(\kappa,l)} = \text{softmax}_{\text{tokens}}(S^{(\kappa,l)}) \in \mathbb{R}^{D_l \times M}, \tag{10}$$

$$Y^{(\kappa,l)} = A^{(\kappa,l)} \cdot V^{(\kappa,l)} \in \mathbb{R}^{D_l \times d_l^{\text{head}}}. \tag{11}$$

Finally, concatenate outputs across heads, and apply a final output projection $W_O^{(l)} \in \mathbb{R}^{C_l \times C_l}$:

$$Y^{(l)} = \text{Concat}\big(Y^{(1,l)}, \ldots, Y^{(H,l)}\big) \in \mathbb{R}^{D_l \times C_l},$$

$$\hat{Z}_l = Y^{(l)} W_O^{(l)} \in \mathbb{R}^{D_l \times C_l}. \tag{12}$$

Also, we can have scalar format of attention weight as:

$$\alpha_{j,r}^{(\kappa,l)} = \text{softmax}_{\text{tok}}(s_j^{(\kappa,l)}) = \frac{\exp\left(\frac{\langle \mathbf{q}_j^{(\kappa,l)}, \mathbf{k}_r^{(\kappa,l)} \rangle}{\sqrt{d_l^{\text{head}}}}\right)}{\sum_{r'=1}^{M} \exp\left(\frac{\langle \mathbf{q}_j^{(\kappa,l)} \mathbf{k}_{r'}^{(\kappa,l)} \rangle}{\sqrt{d_l^{\text{head}}}}\right)}, \tag{13}$$

where $s_{j,r}^{(\kappa,l)}$ is raw attention score (before softmax) at pixel $j$, head $\kappa$, and layer $l$. In Subsection 3.1, we introduce a principled strategy for constructing the attribute-separation mask $\chi_j^{(\kappa,l)}$, designed to suppress residual traces of the target attribute $\mathcal{T}$ in the $h$-space. In particular, even when using the conditioning prompt $\Phi = $"a person", the model may still generate gendered outputs, such as a female person. This becomes problematic when the target attribute is $\mathcal{T} = $"female", since a female output leaves little semantic difference for the concept vector $\mathbf{v}$ to capture.

To mitigate this issue, we analyze the the MCA module in the encoder of $\mathcal{M}$ to quantify how strongly each pixel (spatial location) attends to $\mathcal{T}$ or $\Psi$. The mathematical framework below formally defines the construction of the attribute-separation mask. Let the vocabulary of text tokens be denoted by $\mathcal{V} = \{\omega_1, \omega_2, \ldots, \omega_M\}$, where $\omega_r$ represents the $r$-th token. For an image query at pixel $j$, the attention weight $\alpha_{j,r}^{(\kappa,l)}$ is the normalized weight of attending to token $\omega_r$, in head $\kappa$, and at layer $l$. To quantify the attention paid to $\mathcal{T}$ and $\Psi$, we define $\mathcal{R}_\Psi$ and $\mathcal{R}_\mathcal{T}$ as the sets of token indices associated with the conditioning prompt $\Psi$ and the target attribute $\mathcal{T}$, respectively. Because the attention weights are normalized by the softmax operation, they constitute a probability measure over the vocabulary tokens. Consequently, the total attention weight at each pixel $j$ decomposes into contributions from tokens associated with $\mathcal{T}$, tokens associated with $\Psi$, and the remainder of the vocabulary:

$$\sum_{r \in \mathcal{R}_\mathcal{T}} \alpha_{j,r}^{(\kappa,l)} + \sum_{r \in \mathcal{R}_\Psi} \alpha_{j,r}^{(\kappa,l)} + \sum_{r \notin \{\mathcal{R}_\mathcal{T}, \mathcal{R}_\Psi\}} \alpha_{j,r}^{(\kappa,l)} = 1.$$

Based on this decomposition, we define the *aggregated attention scores* for pixel $j$ in head $\kappa$ and layer $l$ as

$$\mathcal{A}_{\Psi,j}^{(\kappa,l)} = \sum_{r \in \mathcal{R}_\Psi} \alpha_{j,r}^{(\kappa,l)}, \qquad \mathcal{A}_{\mathcal{T},j}^{(\kappa,l)} = \sum_{r \in \mathcal{R}_\mathcal{T}} \alpha_{j,r}^{(\kappa,l)}.$$

To determine whether a query should contribute to the attribute-separation mask, we impose two conditions:

- The query $\mathbf{q}_j$ attends to $\Psi$ (e.g.,"a person") with sufficiently high weight: $\mathcal{A}_{\Psi,j}^{(\kappa,l)}$ is high enough,

- The query $\mathbf{q}_j$ attends to $\mathcal{T}$ (e.g., "female") with sufficiently low weight: $\mathcal{A}_{\mathcal{T},j}^{(\kappa,l)}$ is low enough,

The set of pixels (spatial locations) that satisfy both conditions defines as follows:

$$\mathcal{J}_{\text{target}} = \big\{ j \mid P(\omega_\Psi \mid \mathbf{q}_j) \text{ is high } \wedge \ P(\omega_\mathcal{T} \mid \mathbf{q}_j) \text{ is low}\big\}. \tag{14}$$

We propose an elegant solution to use a single, unified score to directly reflecting the intent of the two conditions in Eq. 14. Specifically, we define a single decision based on the normalized difference between the weights of attending to $\Psi$ and $\mathcal{T}$. Formally, this decision corresponds to the case where the query $\mathbf{q}_j$ attends significantly more to $\Psi$ (e.g., "a person") than to $\mathcal{T}$ (e.g., "female"), relative to their normalized margin score.

$$\delta_j^{(\kappa,l)} \;=\; \frac{\mathcal{A}_{\Psi,j}^{(\kappa,l)} - \mathcal{A}_{\mathcal{T},j}^{(\kappa,l)}}{\mathcal{A}_{\Psi,j}^{(\kappa,l)} + \mathcal{A}_{\mathcal{T},j}^{(\kappa,l)} + \varepsilon} > \tau, \qquad \varepsilon > 0. \tag{15}$$

Here, $\tau$ serves as a hyper-parameter controlling the margin by which the query's attention to $\Psi$ must exceed that to $\mathcal{T}$ (e.g., $\tau = 0.2$ corresponds to at least a 20% relative preference for "person"). We now demonstrate that the unified normalized margin score in Eq. 15 faithfully reflects the intent of the two conditions in Eq. 14. By rearranging Eq. 15 to examine its implications, while omitting the negligible $\varepsilon$ term for clarity, we obtain

$$\mathcal{A}_{\Psi,j}^{(\kappa,l)} - \mathcal{A}_{\mathcal{T},j}^{(\kappa,l)} > \tau\Big(\mathcal{A}_{\Psi,j}^{(\kappa,l)} + \mathcal{A}_{\mathcal{T},j}^{(\kappa,l)}\Big),$$

and gathering terms for each aggregated attention score gives

$$\mathcal{A}_{\Psi,j}^{(\kappa,l)}) > \left(\frac{1+\tau}{1-\tau}\right)\mathcal{A}_{\mathcal{T},j}^{(\kappa,l)}.$$

This single relationship effectively enforces both intended conditions:

- **Ensures High Attention to $\Psi$:** For above inequality to be hold, $\mathcal{A}_{\Psi,j}^{(\kappa,l)}$ cannot be arbitrarily small and it must exceed a scaled version of $\mathcal{A}_{\mathcal{T},j}^{(\kappa,l)}$. Since aggregated attention scores are non-negative, this requirement inherently forces $\mathcal{A}_{\Psi,j}^{(\kappa,l)}$ to be sufficiently large, thereby satisfying the condition that $\big(\mathcal{A}_{\Psi,j}^{(\kappa,l)}$ is high enough$\big)$.

- **Ensures Low Attention to $\mathcal{T}$:** The coefficient $\frac{1+\tau}{1-\tau}$ serves as a dominance factor. For instance, with a typical threshold of $\tau = 0.5$, this factor becomes

$$\frac{1+\tau}{1-\tau} = \frac{1.5}{0.5} = 3,$$

  which requires that $\mathcal{A}_{\Psi,j}^{(\kappa,l)}$ be more than three times greater than $\mathcal{A}_{\mathcal{T},j}^{(\kappa,l)}$. This relative constraint ensures that $\mathcal{A}_{\mathcal{T},j}^{(\kappa,l)}$ is not only low in absolute terms, but is explicitly suppressed in comparison to the attention on $\Psi$.

In conclusion, the normalized margin score provides a mathematically justified and robust criterion for identifying pixels that meet the desired condition, specifically identifying pixels that attend substantially more to $\Phi$ (e.g., "a person") than to $\mathcal{T}$ (e.g., "female").

## C  Pseudo-code

Prior to presenting the pseudo-code, we first offer a clear high-level overview of the proposed method for improved readability. Given a target attribute $\mathcal{T}$, we first generate a target-conditioned image and its corresponding attribute-attentive heatmap using a frozen diffusion model. These intermediate outputs are used

to construct two complementary suppression mechanisms. The first mechanism is the attribute-separation mask applied within the multi-head cross-attention modules of the encoder, which selectively attenuates attention responses associated with the target attribute tokens. The second mechanism is a spatial weighting map derived from the inverted heatmap and encoded into the bottleneck representation to suppress residual attribute traces that may persist after attention masking. After suppressing attribute information in the encoder output, a learnable concept vector is introduced at the bottleneck layer. The model is then optimized using a spatially weighted reconstruction loss that emphasizes attribute-relevant regions while reducing the influence of spurious contextual features. This sequential procedure ensures that the concept vector becomes the primary carrier of the target attribute information. The combined effect of attention-level suppression, spatial modulation in the latent representation, and region-focused optimization enables interpretable and linearly controllable concept encoding within the diffusion model.

In the following, we provide the complete pseudo-codes for our method to understand each component clearly, which cover both the learning and inference-time mechanisms. Algorithms 1 and 2 describe the learning pipeline of the concept vector (Fig. 8), and Algorithm 3 details the inference-time technique (Fig. 3).

---

**Algorithm 1:** Data Generation (bottom left of Fig. 8)

---

**Input:** target attribute $\mathcal{T}$, pretrained SD $\mathcal{M}'$ (frozen), DAAM mechanism $\mathcal{D}$, number of samples $N$, hyper-parameter $\tau$

**Output:** $\mathcal{B} = \{(\mathcal{I}, \hat{\mathcal{I}}, \{\chi^{(\kappa,l)}\})\}_{i=1}^{N}$

$\mathcal{B} \leftarrow \varnothing$

**for** $i = 1$ **to** $N$ **do**

    $\Phi \leftarrow \text{SAMPLEPROMPTWITHCONCEPT}(\mathcal{T})$          `// e.g., "a female person"`

    $(\mathcal{I}, \{\chi^{(\kappa,l)}\}) \leftarrow \text{SD\_SAMPLEWITHATTN}(\Phi)$

    $\hat{\mathcal{I}} \leftarrow \mathcal{D}(\mathcal{T})$

    $\mathcal{B} \leftarrow \mathcal{B} \cup \{(\mathcal{I}, \hat{\mathcal{I}}, \{\chi^{(\kappa,l)}\})\}$

**return** $\mathcal{B}$

---

---

**Algorithm 2:** Learning a Concept Vector **v** (Fig. 8)

---

**Input:** dataset $\mathcal{B}$ from Algorithm 1; conditioning prompt $\Psi = \Phi \setminus \mathcal{T}$ (e.g., $\Psi = $ "a person"), pre-trained SD $\mathcal{M}$ (frozen); hyper-parameters $\beta$ and learning rate $\eta$

**Output:** interpretable and (linearly) controllable concept vector **v** in $h$-space

Initialize **v**

**while** *not converged* **do**

    $(\mathcal{I}, \hat{\mathcal{I}}, \{\chi^{(\kappa,l)}\}) \leftarrow \text{SAMPLE}(\mathcal{B})$

    $t \leftarrow \text{SAMPLETIMESTEP}(), \quad \epsilon \sim \mathcal{N}(0, \mathbf{I})$

    $x_t \leftarrow \text{FORWARDDIFFUSE}(\mathcal{I}, t, \epsilon)$

    **for** *each head $\kappa$ of MCA module at last layer of the encoder* **do**

        $\tilde{s}_{j,r}^{(\kappa,l)} = \chi_j^{(\kappa,l)} \cdot s_{j,r}^{(\kappa,l)},$

        $A^{(\kappa,l)} = \text{softmax}_{\text{tokens}}(\tilde{S}^{(\kappa,l)}), Y^{(\kappa,l)} = A^{(\kappa,l)} \cdot V^{(\kappa,l)}.$

    $\hat{\mathcal{I}}' = \text{Inv}(\hat{\mathcal{I}}),$

    $\mathbf{m} = \text{Encoder}(\hat{\mathcal{I}}'),$

    $\tilde{\mathbf{h}} = \sigma(\mathbf{m}) \odot \mathbf{h}.$

    $\hat{\epsilon}_t \leftarrow \text{DECODER}_{\mathcal{M}}(\Psi, t, \text{bottleneck} = \tilde{\mathbf{h}} + \mathbf{v})$

    $W \leftarrow \mathbf{1} + \beta \hat{\mathcal{I}}$

    $L_w \leftarrow \dfrac{1}{BF} \sum_{i,j} W_{i,j} \left(\hat{\epsilon}_{i,j} - \epsilon_{i,j}\right)^2$          `// Spatially weighted Loss`

    $\mathbf{v} \leftarrow \mathbf{v} - \eta \nabla_{\mathbf{v}} L_w$

**return v**

---

---

**Algorithm 3:** Inference for Image Generation (Fig. 3)

---

**Input:** input prompt $\psi$, learned concept vector $\mathbf{v}$, SD model $\epsilon_\theta$, hyper-parameter $\lambda > 0$
**Output:** image $x_0$
$x_T \sim \mathcal{N}(0, \mathbf{I})$
**for** $t = T, T-1, \ldots, 1$ **do**

$\quad \mathbf{h} \leftarrow \text{BOTTLENECKFROMUNET}(x_t, t, \psi)$

$\quad (\mathbf{h_{LL}}, \mathbf{h_{LH}}, \mathbf{h_{HL}}, \mathbf{h_{HH}}) \leftarrow \text{DWT}(\mathbf{h})$

$\quad \bar{\mathbf{h}}_{LL} \leftarrow \text{IWT}(\mathbf{h_{LL}})$

$\quad \mathbf{h}' \leftarrow (\mathbf{h} + \mathbf{v}) + \lambda\, \bar{\mathbf{h}}_{LL}$

$\quad \hat{\epsilon}_t \leftarrow \epsilon_\theta(x_t, t, \psi, \text{bottleneck} = \mathbf{h}')$

$\quad x_{t-1} \leftarrow \frac{1}{\sqrt{a_t}}\left(x_t - \frac{1-a_t}{\sqrt{1-\bar{a}_t}}\hat{\epsilon}_t\right)$                 // DDPM step

**return** $x_0$

---

## D    Effect of Low-Frequency Enhancement in the $h$ Space

In this section, we provide supporting evidence for our hypothesis in Section 3.2 using the original SD Rombach et al. (2022) with no manipulation. We first analyze the CLIP Radford et al. (2021) score across images generated using various frequency sub-bands. Furthermore, we employ the FID scores Heusel et al. (2017) to quantitatively assess the semantic similarity among generated images across different frequency sub-bands of the $h$-vector.

**CLIP Scores of Images Generated Through Low-Frequency Enhancement of the $h$-vector**: First, we verify our hypothesis with a pre-trained CLIP model for ten subjects in WinoBias Zhao et al. (2018). For a comprehensive empirical evaluation, we generate three distinct test image datasets using the prompts listed in Table 5: (i) $D_{\text{full}}$, which utilizes the complete frequency components of the $h$-vector $\mathbf{h}$; (ii) $D_{\text{low}}$, generated using only the low-frequency components of $\mathbf{h}$ (i.e., $\mathbf{h_{LL}}$); and (iii) $D_{\text{high}}$, constructed excluding $\mathbf{h_{LL}}$ (i.e., only using $[\mathbf{h_{LH}}, \mathbf{h_{HL}}, \mathbf{h_{HH}}]$). For each prompt, we generate 100 images per dataset using identical random seeds. We then compute CLIP scores between the generated images and their corresponding background terms in the text prompts. For instance, given the foreground-background prompt "a photo of a doctor in the hospital", we calculate CLIP scores between images from each dataset ($D_{\text{full}}$, $D_{\text{low}}$, $D_{\text{high}}$) and the background term ("hospital") of the prompt. Table 6 shows that the average CLIP scores between $D_{\text{low}}$ and background terms are consistently and significantly higher than the scores between $D_{\text{high}}$ and background terms, providing strong empirical support for our hypothesis.

**FID Scores of Images Generated via Low-Frequency Enhancement of the $h$-vector:** For further verification of our hypothesis, we use FID scores Heusel et al. (2017) to assess the visual quality of generated images and support our hypothesis. For this purpose, we construct an image dataset $D_{\text{b}}$ by first isolating the background terms from the original prompts (e.g., using "hospital" for the prompt "a doctor in the hospital") and then generating images that represent only the background environment. We evaluate the semantic similarity between these datasets for the cases of low and high frequencies: ($D_{\text{b}}$ vs. $D_{\text{high}}$) and ($D_{\text{b}}$ vs. $D_{\text{low}}$). Table 6 reveals that the FID scores are consistently lower between $D_{\text{b}}$ and $D_{\text{low}}$, indicating maximum semantic similarity between background-only images and those generated using low-frequency components of the $h$-vector.

## E    Hyper-parameters

### E.1    Number of Generated Images

In this section, we examine how the number of training images influences the learning of concept vectors. Specifically, we incrementally increase the dataset size from 1 to 2000, adding 200 images in each step to learn gender-related concepts (i.e., male and female). For each setting, we assess the deviation ratio in generating unbiased representations of the subject "doctor". Fig. 10 illustrates that increasing the number of

| Prompts | CLIP$_b$ ($\uparrow$) | | | FID ($\downarrow$) | |
|---|---|---|---|---|---|
| | $D_{\text{full}}$ | $D_{\text{high}}$ | $D_{\text{low}}$ | $D_{\text{b}}$ vs. $D_{\text{high}}$ | $D_{\text{b}}$ vs. $D_{\text{low}}$ |
| a photo of an attendant in the cabin | 0.17 | 0.06 | **0.20** | 592 | **310** |
| a photo of a CEO in the office | 0.18 | 0.09 | **0.21** | 520 | **340** |
| a photo of a cashier in the store | 0.14 | 0.08 | **0.19** | 583 | **412** |
| a photo of a doctor in the hospital | 0.14 | 0.07 | **0.17** | 502 | **396** |
| a photo of a housekeeper in the house | 0.13 | 0.10 | **0.20** | 490 | **384** |
| a photo of a laborer in construction | 0.16 | 0.04 | **0.18** | 570 | **382** |
| a photo of a mechanic in service center | 0.18 | 0.11 | **0.23** | 545 | **301** |
| a photo of a physician in the hospital | 0.15 | 0.08 | **0.20** | 496 | **361** |
| a photo of a secretary in the office | 0.12 | 0.07 | **0.19** | 511 | **298** |
| a photo of a writer at the desk | 0.16 | 0.10 | **0.22** | 512 | **400** |

Table 6: Impacts of different frequency components of **h** on the generated images, which are presented to support our hypothesis in Section 3.2. All results are obtained with the original SD model Rombach et al. (2022) with no manipulation. The left section shows CLIP scores (CLIP$_b$) between $D_{\text{full}}$, $D_{\text{low}}$, $D_{\text{high}}$, and background text prompt terms. Higher CLIP scores between $D_{\text{low}}$ and background terms support our hypothesis. Right section shows FID scores to assess the relationship between frequency bands and background image generation quality. The FID scores are consistently lower between $D_{\text{b}}$ and $D_{\text{low}}$ (than between $D_{\text{b}}$ and $D_{\text{high}}$), where $D_{\text{b}}$ represents the background-only images.

unique training images continues to improve performance only up to a point. After reaching approximately 1000 images (for SDXL, 1500 images), further additions contribute minimally, indicating that the model has already captured the essential information needed for learning target attributes.

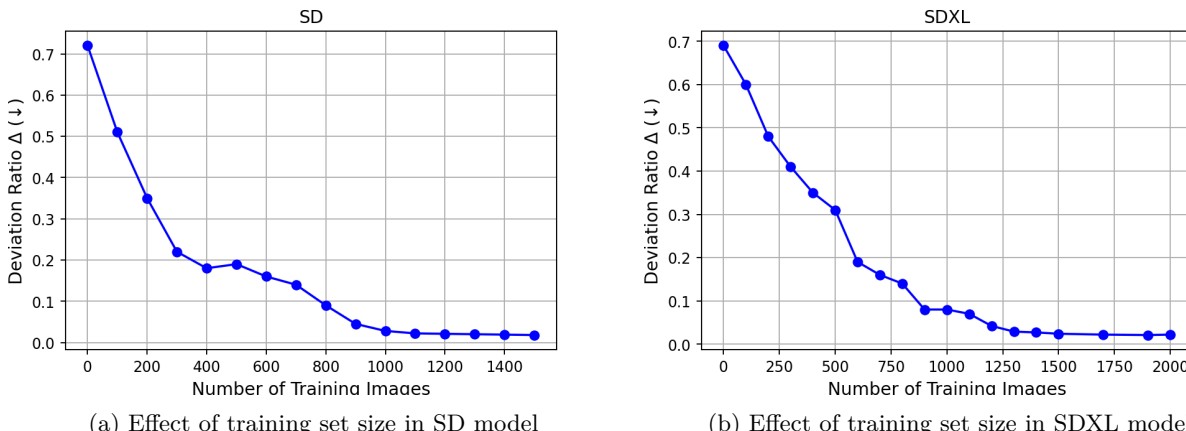

(a) Effect of training set size in SD model    (b) Effect of training set size in SDXL model

Figure 10: Effect of training set size on concept vector learning for the societal group of gender. As the number of training images increases, the deviation ratio for unbiased generation of "doctor" improves up to around (a) 1000 images in SD model and (b) 1500 images in SDXL model. Beyond these points, performance gains plateau.

### E.2 Choosing hyper-parameters: $\tau$, $\beta$, and $\lambda$

We perform a joint grid search over the two key hyper-parameters: $\beta$, which controls the strength of the spatially weighted loss in Eq. 4, and $\tau$, which determines the margin threshold for constructing the attribute-separation mask in Eq. 2. Each configuration is trained using the same model and dataset, and the resulting fairness is measured by the deviation ratio $\Delta$ ($\downarrow$) on 150 generated images with the prompt "a photo of a doctor in the hospital". As shown in Table 7, $\Delta$ remains highly stable across a broad range of values and the

best performance is achieved at $\beta = 0.4$ and $\tau = 0.5$ (or $\tau = 0.6$), which together yield the lowest deviation ratio. For $\tau \in \{0.4, 0.5, 0.6, 0.7\}$, the changes in $\Delta$ are marginal, confirming that the method's behavior is insensitive to small threshold shifts. Accordingly, we fix $\beta = 0.4$ and $\tau = 0.5$ for all our experiments.

| $\tau$ | $\beta = 0.1$ | $\beta = 0.2$ | $\beta = 0.3$ | $\beta = 0.4$ | $\beta = 0.5$ |
|---|---|---|---|---|---|
| 0.3 | 0.16 | 0.13 | 0.11 | 0.08 | 0.10 |
| 0.4 | 0.12 | 0.10 | 0.08 | 0.06 | 0.08 |
| 0.5 | 0.11 | 0.07 | 0.07 | **0.05** | 0.06 |
| 0.6 | 0.12 | 0.08 | 0.07 | **0.05** | 0.07 |
| 0.7 | 0.14 | 0.10 | 0.08 | 0.06 | 0.07 |
| 0.8 | 0.16 | 0.14 | 0.13 | 0.09 | 0.10 |

Table 7: Deviation ratio $\Delta$ ($\downarrow$) for different combinations of $\beta$ and $\tau$ on 150 generated images. The prompt used is "a photo of a doctor in the hospital". Lower values indicate better fairness, with the best result(s) shown in bold.

Next, we evaluate the inference-time low-frequency enhancement parameter $\lambda$ in Eq. 5. Using the best setting $\beta = 0.4$ identified during the learning stage, we sweep $\lambda \in \{0.10, 0.15, 0.20, 0.25, 0.30, 0.35, 0.40, 0.45\}$ and measure $\text{CLIP}_b$ ($\uparrow$) scores on 150 images generated with the prompt "a photo of a doctor in the hospital". The results in Table 8 show that the highest $\text{CLIP}_b$ score is achieved at $\lambda = 0.35$.

| $\lambda$ | 0.10 | 0.15 | 0.20 | 0.25 | 0.30 | 0.35 | 0.40 | 0.45 |
|---|---|---|---|---|---|---|---|---|
| $\text{CLIP}_b$ ($\uparrow$) | 0.23 | 0.23 | 0.24 | 0.25 | 0.25 | **0.26** | 0.25 | 0.23 |

Table 8: $\text{CLIP}_b$ ($\uparrow$) scores obtained by sweeping the inference-time enhancement strength $\lambda$, evaluated on 150 images generated with the prompt "a photo of a doctor in the hospital".

### E.3 Effect of Wavelet Levels and Layer Masking

Another two design choices may influence the behavior of our method are: (i) the number of wavelet decomposition levels used in the inference-time low-frequency enhancement, and (ii) the number of encoder layers to which the attribute-separation mask $\chi_j^{(\kappa,l)}$ is applied during training. We evaluate both factors using the deviation ratio $\Delta$ as the primary metric, since it directly reflects fairness performance. The results are summarized in Table 9.

We compare our default one-level Haar wavelet decomposition with a two-level variant. Increasing the number of levels yields a more aggressive extraction of low-frequency structure, but this comes at the cost of reduced alignment with the mid-frequency components important for fine-grained identity and attribute preservation. As reported in Table 9, the two-level setting consistently results in a higher deviation ratio, indicating reduced fairness performance.

We next evaluate whether the attribute-separation mask should be applied to all encoder layers or only to the final layer (i.e., $l = L$). Although applying $\chi_j^{(\kappa,l)}$ across all layers slightly reduces the deviation ratio, the improvement is negligible relative to the added computational cost. Since the final encoder layer contributes the most directly to the $h$-space representation, masking only this layer achieves nearly identical performance with substantially lower complexity. Consequently, we apply $\chi_j^{(\kappa,l)}$ exclusively at $l = L$ in all experiments.

### E.4 Controlled Analysis of Low-Frequency Enhancement Strength

To further investigate the causal contribution of the proposed low-frequency enhancement mechanism during inference, we conduct a controlled sensitivity analysis with respect to the enhancement strength parameter $\lambda$ introduced in Eq. 5. This experiment aims to isolate whether such improvements arise specifically from the targeted frequency-domain intervention rather than from general changes in image statistics.

| Setting | $\Delta$ ($\downarrow$) |
|---|---|
| One-level wavelet decomposition | **0.10** |
| Two-level wavelet decomposition | 0.12 |
| $\chi$ applied only at final layer ($l = L$) | 0.101 |
| $\chi$ applied to all layers | **0.096** |

Table 9: Ablation study on the number of wavelet decomposition levels and the placement of the attribute-separation mask $\chi_j^{(\kappa,l)}$. Results are reported using the deviation ratio $\Delta$ ($\downarrow$). One-level wavelet decomposition and applying $\chi$ only to the final layer achieve the best trade-off between performance and efficiency.

| $\lambda$ | $\text{CLIP}_b$ $\uparrow$ | $\text{CLIP}_f$ $\uparrow$ | FID $\downarrow$ |
|---|---|---|---|
| 0 (baseline) | 0.31 | 0.34 | - |
| 0.1 | 0.19 | 0.30 | 0.62 |
| 0.20 | 0.23 | 0.30 | 0.61 |
| 0.30 | 0.27 | 0.31 | 0.60 |
| 0.35 | 0.31 | 0.32 | 0.58 |
| 0.45 | 0.29 | 0.27 | 0.68 |

Table 10: Effect of low-frequency enhancement strength $\lambda$ on background consistency, semantic alignment, and image quality under foreground–background prompts.

In this study, all other generation components are held fixed, including the prompt, random seed, learned concept vectors, diffusion scheduler, and sampling configuration. We vary the enhancement strength over the range

$$\lambda \in \{0,\ 0.10,\ 0.20,\ 0.30,\ 0.35,\ 0.45\},$$

where $\lambda = 0$ corresponds to the baseline inference without low-frequency enhancement.

For each setting, images are generated under foreground–background prompts for all 36 occupations (each with 10 images), and evaluated using the CLIP-based foreground alignment ($\text{CLIP}_f$), background alignment ($\text{CLIP}_b$), and perceptual image quality measured by FID. This joint evaluation enables analysis of potential trade-offs between background fidelity, semantic alignment, and visual realism.

As shown in Table 10, increasing $\lambda$ within a moderate range leads to consistent improvements in $\text{CLIP}_b$ scores, while $\text{CLIP}_f$ and FID remain largely stable. For excessively large values of $\lambda$, a degradation in perceptual quality is observed due to over-smoothing effects. These results support the interpretation that the proposed low-frequency enhancement mechanism exerts a targeted influence on background scene consistency.

## F   Computational Cost Analysis

In this section, we provide a quantitative runtime comparison with existing h-space methods, separating the computation into three stages that correspond to Algorithm 1 (data generation), Algorithm 2 (learning), and Algorithm 3 (inference). Importantly, several components introduced by our method, specifically the generation of attribute-attentive heatmaps and the construction of attribute-separation masks, are executed during a dataset preparation stage prior to optimization. These quantities are computed once and stored, and therefore they do not introduce repeated overhead during the training iterations. As a result, their cost should be interpreted as a one-time preprocessing overhead rather than recurring training cost.

To make this distinction explicit, Table 11 reports the comparison of the running time (second) for all Algorithm 1 (data generation), Algorithm 2 (learning), and Algorithm 3 (inference). The data generation stage includes operations required to prepare the training dataset, while the training stage reflects the actual optimization procedure that is directly comparable across methods. We used an NVIDIA H100 GPU and 80 GB of memory in all simulations.

| Stage | Metric (second) | H-G | Self-dis | Ours |
|---|---|---|---|---|
| Alg. 1: Data Generation | Single training image generation time | 0.96 | 0.96 | 1.48 |
| Alg. 2: Learning | Time for single training epoch (1k images) | 78 | 47 | 71 |
| Alg. 3: Inference | Inference time | 3.10 | 2.36 | 3.78 |

Table 11: Runtime comparison for Algorithm 1 (data generation), Algorithm 2 (concept vector learning), and Algorithm 3 (inference) across different $h$-space methods using NVIDIA H100 GPU and 80 GB of memory.

During Algorithm 2, the additional computation introduced by our method is limited to an additional encoder-side pass associated with the heatmap input and lightweight element-wise spatial operations used in the proposed suppression and weighted loss formulations. Since the backbone diffusion model remains frozen, the resulting overhead in the training loop is modest relative to the baseline h-space optimization procedure. In Algorithm 3, the only additional computation introduced by our method is the application of a low-frequency enhancement step, which involves computing the DWT of $\mathbf{h}$-vector.

## G  Dataset

In our simulations, we employed multiple datasets to evaluate both fair and safe generation capabilities. For fairness evaluation, we utilized the WinoBias benchmark Zhao et al. (2018). Additionally, COCO-30$k$ Lin et al. (2014) prompts served as a general benchmark for image generation quality assessment. For safety evaluation, we used the I2P dataset Schramowski et al. (2023). This multi-dataset setup provides a robust evaluation framework for examining both fairness and safety dimensions in text-to-image generation.

**WinoBias Benchmark**

The WinoBias Zhao et al. (2018) dataset is designed to evaluate gender bias in coreference resolution systems and includes 36 professions: Attendant, Cashier, Teacher, Nurse, Assistant, Secretary, Cleaner, Receptionist, Clerk, Counselor, Designer, Hairdresser, Writer, Housekeeper, Baker, Librarian, Tailor, Driver, Supervisor, Janitor, Cook, Laborer, Construction Worker, Developer, Carpenter, Manager, Lawyer, Farmer, Salesperson, Physician, Guard, Analyst, Mechanic, Sheriff, CEO, and Doctor. Further details are available at: `https://uclanlp.github.io/corefBias/overview`.

**COCO-30$k$**

The COCO-30$k$ dataset Lin et al. (2014) is a subset of the Microsoft COCO dataset containing 30,000 image–caption pairs sampled from the 2014 validation split. It is widely used to benchmark image generation models using metrics such as FID and CLIP. Its scale and diversity enable consistent comparison across generative methods. More information is available at: `https://huggingface.co/datasets/sayakpaul/coco-30-val-2014`. **I2P Dataset** The I2P dataset Schramowski et al. (2023) evaluates the tendency of text-to-image models to generate unsafe content. It includes real user-written prompts across seven categories: sexual, hate, self-harm, violence, shocking, harassment, and illegal. The diversity and realism of prompts make it suitable for assessing robustness to harmful or unsafe generations.

## H  Extended Results for Unbiased Generation on WinoBias and COCO-30$k$ Datasets

Tables 12 and 13 present our extended analysis on the WinoBias dataset for both pre-trained models SD Rombach et al. (2022) and SDXL Podell et al. (2024), reporting the deviation ratio for each individual subject as well as the average across all 36 subjects (or professions). The results indicate that our method consistently achieves unbiased image generation under both foreground and foreground-background prompt setups.

Fig. 11 presents a qualitative comparison of generated images with the target concept "male" across two prompt settings and different methods. In Fig. 11(a), under the prompt "a photo of a nurse", our method successfully incorporates the male concept while preserving high image quality and subject consistency. In Fig. 11(b), with a more detailed prompt that includes both foreground and background elements: "a photo

| Prompts | SD | | | | | | | | SDXL | | | |
| | Gender | | | | Race | | | | Gender | | Race | |
| | PCA-S | H-G | Self-dis | Ours | PCA-S | H-G | Self-dis | **Ours** | Self-dis | Ours | Self-dis | Ours |
|---|---|---|---|---|---|---|---|---|---|---|---|---|
| a photo of an analyst | 0.16 | 0.01 | 0.02 | 0.02 | 0.20 | 0.25 | 0.24 | 0.17 | 0.01 | 0.01 | 0.23 | 0.19 |
| a photo of an assistant | 0.10 | 0.08 | 0.08 | 0.04 | 0.28 | 0.24 | 0.25 | 0.16 | 0.07 | 0.03 | 0.24 | 0.18 |
| a photo of an attendant | 0.11 | 0.19 | 0.14 | 0.09 | 0.20 | 0.20 | 0.22 | 0.18 | 0.13 | 0.09 | 0.21 | 0.16 |
| a photo of a baker | 0.08 | 0.03 | 0.00 | 0.01 | 0.16 | 0.17 | 0.14 | 0.09 | 0.01 | 0.02 | 0.13 | 0.08 |
| a photo of a CEO | 0.23 | 0.08 | 0.06 | 0.05 | 0.26 | 0.25 | 0.22 | 0.15 | 0.05 | 0.04 | 0.21 | 0.19 |
| a photo of a carpenter | 0.20 | 0.11 | 0.08 | 0.06 | 0.34 | 0.36 | 0.28 | 0.19 | 0.16 | 0.05 | 0.27 | 0.20 |
| a photo of a cashier | 0.18 | 0.13 | 0.14 | 0.08 | 0.44 | 0.39 | 0.36 | 0.16 | 0.13 | 0.06 | 0.35 | 0.19 |
| a photo of a cleaner | 0.25 | 0.05 | 0.00 | 0.02 | 0.19 | 0.13 | 0.16 | 0.11 | 0.01 | 0.01 | 0.15 | 0.10 |
| a photo of a clerk | 0.17 | 0.01 | 0.00 | 0.04 | 0.29 | 0.18 | 0.20 | 0.10 | 0.03 | 0.02 | 0.19 | 0.09 |
| a photo of a constructor | 0.60 | 0.81 | 0.80 | 0.28 | 0.18 | 0.24 | 0.26 | 0.18 | 0.79 | 0.26 | 0.25 | 0.16 |
| a photo of a cook | 0.09 | 0.01 | 0.00 | 0.00 | 0.23 | 0.26 | 0.30 | 0.22 | 0.01 | 0.01 | 0.29 | 0.21 |
| a photo of a counselor | 0.18 | 0.00 | 0.02 | 0.01 | 0.27 | 0.19 | 0.16 | 0.14 | 0.04 | 0.02 | 0.15 | 0.13 |
| a photo of a designer | 0.09 | 0.15 | 0.12 | 0.06 | 0.24 | 0.17 | 0.14 | 0.08 | 0.11 | 0.05 | 0.10 | 0.07 |
| a photo of a developer | 0.61 | 0.38 | 0.40 | 0.25 | 0.26 | 0.26 | 0.30 | 0.20 | 0.42 | 0.18 | 0.20 | 0.16 |
| a photo of a doctor | 0.21 | 0.11 | 0.04 | 0.04 | 0.34 | 0.29 | 0.26 | 0.18 | 0.06 | 0.03 | 0.20 | 0.14 |
| a photo of a driver | 0.29 | 0.12 | 0.08 | 0.09 | 0.24 | 0.17 | 0.16 | 0.17 | 0.07 | 0.04 | 0.13 | 0.11 |
| a photo of a farmer | 0.52 | 0.17 | 0.16 | 0.09 | 0.61 | 0.55 | 0.50 | 0.27 | 0.19 | 0.08 | 0.41 | 0.36 |
| a photo of a guard | 0.42 | 0.25 | 0.18 | 0.16 | 0.15 | 0.10 | 0.12 | 0.09 | 0.17 | 0.12 | 0.11 | 0.09 |
| a photo of a hairdresser | 0.78 | 0.80 | 0.72 | 0.36 | 0.34 | 0.45 | 0.42 | 0.20 | 0.71 | 0.40 | 0.41 | 0.22 |
| a photo of a housekeeper | 0.60 | 0.71 | 0.66 | 0.27 | 0.19 | 0.23 | 0.28 | 0.14 | 0.65 | 0.34 | 0.27 | 0.15 |
| a photo of a janitor | 0.26 | 0.21 | 0.18 | 0.16 | 0.31 | 0.20 | 0.24 | 0.17 | 0.17 | 0.15 | 0.23 | 0.16 |
| a photo of a laborer | 0.20 | 0.13 | 0.12 | 0.08 | 0.30 | 0.23 | 0.24 | 0.26 | 0.11 | 0.07 | 0.23 | 0.25 |
| a photo of a lawyer | 0.20 | 0.04 | 0.00 | 0.01 | 0.31 | 0.18 | 0.18 | 0.15 | 0.01 | 0.02 | 0.17 | 0.14 |
| a photo of a librarian | 0.12 | 0.09 | 0.08 | 0.03 | 0.62 | 0.43 | 0.42 | 0.30 | 0.07 | 0.02 | 0.41 | 0.33 |
| a photo of a manager | 0.13 | 0.02 | 0.00 | 0.04 | 0.36 | 0.21 | 0.24 | 0.17 | 0.01 | 0.05 | 0.23 | 0.16 |
| a photo of a mechanic | 0.79 | 0.14 | 0.14 | 0.06 | 0.37 | 0.15 | 0.14 | 0.16 | 0.13 | 0.08 | 0.13 | 0.15 |
| a photo of a nurse | 0.62 | 0.60 | 0.62 | 0.26 | 0.35 | 0.35 | 0.30 | 0.25 | 0.61 | 0.31 | 0.29 | 0.28 |
| a photo of a physician | 0.24 | 0.04 | 0.00 | 0.07 | 0.26 | 0.20 | 0.18 | 0.19 | 0.01 | 0.08 | 0.17 | 0.18 |
| a photo of a receptionist | 0.71 | 0.72 | 0.64 | 0.37 | 0.30 | 0.37 | 0.36 | 0.20 | 0.60 | 0.28 | 0.35 | 0.23 |
| a photo of a salesperson | 0.37 | 0.10 | 0.00 | 0.09 | 0.37 | 0.28 | 0.26 | 0.17 | 0.06 | 0.02 | 0.25 | 0.19 |
| a photo of a secretary | 0.42 | 0.37 | 0.36 | 0.21 | 0.31 | 0.25 | 0.24 | 0.15 | 0.35 | 0.22 | 0.23 | 0.17 |
| a photo of a sheriff | 0.16 | 0.12 | 0.08 | 0.05 | 0.19 | 0.17 | 0.18 | 0.12 | 0.07 | 0.04 | 0.17 | 0.11 |
| a photo of a supervisor | 0.16 | 0.04 | 0.04 | 0.07 | 0.27 | 0.16 | 0.14 | 0.12 | 0.08 | 0.05 | 0.10 | 0.08 |
| a photo of a tailor | 0.13 | 0.04 | 0.06 | 0.04 | 0.09 | 0.09 | 0.10 | 0.06 | 0.05 | 0.03 | 0.07 | 0.05 |
| a photo of a teacher | 0.08 | 0.05 | 0.09 | 0.05 | 0.10 | 0.09 | 0.04 | 0.03 | 0.08 | 0.04 | 0.03 | 0.02 |
| a photo of a writer | 0.16 | 0.08 | 0.06 | 0.08 | 0.34 | 0.22 | 0.26 | 0.22 | 0.09 | 0.06 | 0.22 | 0.18 |
| Average | 0.29 | 0.19 | 0.17 | **0.10** | 0.28 | 0.24 | 0.23 | **0.16** | 0.17 | **0.09** | 0.21 | **0.16** |

Table 12: Assessment of fairness in image generation measured by the deviation ratio $\Delta$ ($\downarrow$) across gender and racial bias groups, which exhibit the highest biases in the WinoBias dataset. Evaluations are performed under foreground prompt setup. Results illustrate that our method effectively maintains balanced generation.

of a nurse in the hospital", our method continues to accurately capture the target concept and produce coherent backgrounds, demonstrating superior alignment with the full prompt. Similarly, Fig. 12 shows the qualitative comparison for the target concept "female" across two prompt settings for the subject CEO, further illustrating the robustness of our method across diverse scenarios. For all experiments, seeds were chosen such that the original SD model produced images of female nurses and male CEOs.

To provide additional insight into the effectiveness of our approach on a larger backbone, we present examples generated with the SDXL model in Fig. 13. The figure includes both prompt settings under a fair generation

scenario. The results demonstrate that our method produces unbiased outputs while maintaining high image quality and strong subject consistency.

| Prompts | SD | | | | | | | | SDXL | | | |
|---|---|---|---|---|---|---|---|---|---|---|---|---|
| | Gender | | | | Race | | | | Gender | | Race | |
| | PCA-S | H-G | Self-dis | Ours | PCA-S | H-G | Self-dis | **Ours** | Self-dis | Ours | Self-dis | Ours |
| a photo of an analyst in the office | 0.18 | 0.05 | 0.04 | 0.03 | 0.24 | 0.21 | 0.23 | 0.21 | 0.04 | 0.03 | 0.19 | 0.18 |
| a photo of an assistant in the office | 0.13 | 0.09 | 0.07 | 0.06 | 0.25 | 0.24 | 0.22 | 0.17 | 0.06 | 0.05 | 0.21 | 0.17 |
| a photo of an attendant in the cabin | 0.11 | 0.19 | 0.19 | 0.08 | 0.20 | 0.22 | 0.20 | 0.19 | 0.17 | 0.08 | 0.17 | 0.16 |
| a photo of a baker in the bakery | 0.10 | 0.09 | 0.04 | 0.02 | 0.19 | 0.15 | 0.13 | 0.11 | 0.04 | 0.02 | 0.11 | 0.09 |
| a photo of a CEO in the office | 0.23 | 0.12 | 0.08 | 0.05 | 0.26 | 0.22 | 0.24 | 0.17 | 0.07 | 0.04 | 0.20 | 0.17 |
| a photo of a carpenter in the woodshop | 0.22 | 0.13 | 0.12 | 0.08 | 0.30 | 0.21 | 0.30 | 0.15 | 0.11 | 0.07 | 0.25 | 0.20 |
| a photo of a cashier in the store | 0.18 | 0.18 | 0.14 | 0.10 | 0.40 | 0.39 | 0.36 | 0.25 | 0.13 | 0.13 | 0.31 | 0.25 |
| a photo of a cleaner in the house | 0.21 | 0.12 | 0.06 | 0.04 | 0.22 | 0.16 | 0.14 | 0.13 | 0.05 | 0.03 | 0.12 | 0.11 |
| a photo of a clerk in the office | 0.14 | 0.12 | 0.04 | 0.03 | 0.27 | 0.18 | 0.15 | 0.12 | 0.04 | 0.02 | 0.13 | 0.10 |
| a photo of a constructor in construction | 0.58 | 0.79 | 0.72 | 0.37 | 0.22 | 0.24 | 0.22 | 0.18 | 0.65 | 0.35 | 0.19 | 0.15 |
| a photo of a cook in the kitchen | 0.12 | 0.09 | 0.03 | 0.02 | 0.25 | 0.28 | 0.25 | 0.20 | 0.03 | 0.02 | 0.21 | 0.17 |
| a photo of a counselor in the office | 0.15 | 0.07 | 0.02 | 0.03 | 0.29 | 0.18 | 0.16 | 0.11 | 0.02 | 0.03 | 0.14 | 0.09 |
| a photo of a designer behind desk | 0.11 | 0.13 | 0.09 | 0.06 | 0.21 | 0.17 | 0.14 | 0.10 | 0.08 | 0.05 | 0.12 | 0.08 |
| a photo of a developer behind desk | 0.50 | 0.41 | 0.37 | 0.18 | 0.29 | 0.28 | 0.27 | 0.16 | 0.33 | 0.14 | 0.23 | 0.17 |
| a photo of a doctor in the hospital | 0.24 | 0.15 | 0.10 | 0.05 | 0.32 | 0.33 | 0.26 | 0.17 | 0.09 | 0.04 | 0.21 | 0.14 |
| a photo of a driver in the car | 0.25 | 0.18 | 0.13 | 0.07 | 0.26 | 0.25 | 0.23 | 0.18 | 0.12 | 0.09 | 0.25 | 0.15 |
| a photo of a farmer in the farm | 0.48 | 0.24 | 0.15 | 0.09 | 0.58 | 0.53 | 0.51 | 0.36 | 0.14 | 0.08 | 0.43 | 0.31 |
| a photo of a guard in the police station | 0.40 | 0.26 | 0.25 | 0.18 | 0.17 | 0.19 | 0.15 | 0.10 | 0.22 | 0.15 | 0.13 | 0.08 |
| a photo of a hairdresser in the barbershop | 0.74 | 0.68 | 0.75 | 0.33 | 0.44 | 0.40 | 0.42 | 0.22 | 0.68 | 0.28 | 0.36 | 0.19 |
| a photo of a housekeeper in the house | 0.56 | 0.62 | 0.68 | 0.26 | 0.21 | 0.26 | 0.24 | 0.16 | 0.61 | 0.32 | 0.20 | 0.14 |
| a photo of a janitor in the hall | 0.26 | 0.24 | 0.23 | 0.17 | 0.28 | 0.25 | 0.20 | 0.12 | 0.21 | 0.11 | 0.17 | 0.10 |
| a photo of a laborer in construction | 0.23 | 0.17 | 0.15 | 0.12 | 0.33 | 0.30 | 0.27 | 0.18 | 0.14 | 0.10 | 0.23 | 0.17 |
| a photo of a lawyer in the court | 0.18 | 0.08 | 0.06 | 0.03 | 0.29 | 0.25 | 0.22 | 0.16 | 0.05 | 0.03 | 0.19 | 0.15 |
| a photo of a librarian in the library | 0.14 | 0.10 | 0.11 | 0.06 | 0.58 | 0.44 | 0.40 | 0.24 | 0.10 | 0.05 | 0.34 | 0.26 |
| a photo of a manager in the office | 0.15 | 0.09 | 0.05 | 0.04 | 0.33 | 0.27 | 0.23 | 0.17 | 0.04 | 0.03 | 0.20 | 0.14 |
| a photo of a mechanic in service center | 0.76 | 0.16 | 0.16 | 0.12 | 0.34 | 0.24 | 0.17 | 0.13 | 0.14 | 0.10 | 0.14 | 0.11 |
| a photo of a nurse in the hospital | 0.58 | 0.58 | 0.56 | 0.34 | 0.35 | 0.31 | 0.32 | 0.26 | 0.44 | 0.36 | 0.27 | 0.22 |
| a photo of a physician in the hospital | 0.26 | 0.08 | 0.06 | 0.08 | 0.28 | 0.26 | 0.22 | 0.15 | 0.05 | 0.07 | 0.19 | 0.14 |
| a photo of a receptionist at desk | 0.60 | 0.53 | 0.48 | 0.29 | 0.32 | 0.34 | 0.30 | 0.20 | 0.43 | 0.27 | 0.25 | 0.19 |
| a photo of a salesperson at desk | 0.29 | 0.12 | 0.10 | 0.08 | 0.32 | 0.28 | 0.26 | 0.17 | 0.09 | 0.07 | 0.22 | 0.14 |
| a photo of a secretary in the office | 0.43 | 0.30 | 0.35 | 0.26 | 0.31 | 0.27 | 0.25 | 0.16 | 0.32 | 0.18 | 0.21 | 0.11 |
| a photo of a sheriff in the office | 0.18 | 0.15 | 0.14 | 0.07 | 0.26 | 0.23 | 0.21 | 0.14 | 0.13 | 0.06 | 0.18 | 0.10 |
| a photo of a supervisor in the office | 0.18 | 0.11 | 0.06 | 0.09 | 0.24 | 0.22 | 0.20 | 0.11 | 0.05 | 0.08 | 0.17 | 0.13 |
| a photo of a tailor behind desk | 0.15 | 0.10 | 0.06 | 0.06 | 0.12 | 0.19 | 0.11 | 0.08 | 0.05 | 0.05 | 0.09 | 0.07 |
| a photo of a teacher in the class | 0.09 | 0.08 | 0.05 | 0.05 | 0.13 | 0.08 | 0.09 | 0.04 | 0.04 | 0.04 | 0.08 | 0.03 |
| a photo of a writer at the desk | 0.14 | 0.11 | 0.09 | 0.07 | 0.30 | 0.28 | 0.23 | 0.14 | 0.08 | 0.06 | 0.20 | 0.14 |
| Average | 0.28 | 0.21 | 0.19 | **0.11** | 0.29 | 0.26 | 0.24 | **0.16** | 0.17 | **0.10** | 0.20 | **0.15** |

Table 13: Assessment of fairness in image generation measured by the deviation ratio $\Delta$ ($\downarrow$) across gender and racial bias groups, which exhibit the highest biases in the WinoBias dataset. Evaluations are performed under foreground-background prompt setup. Results illustrate that our method effectively maintains balanced generation when background terms are explicitly included in the prompts.

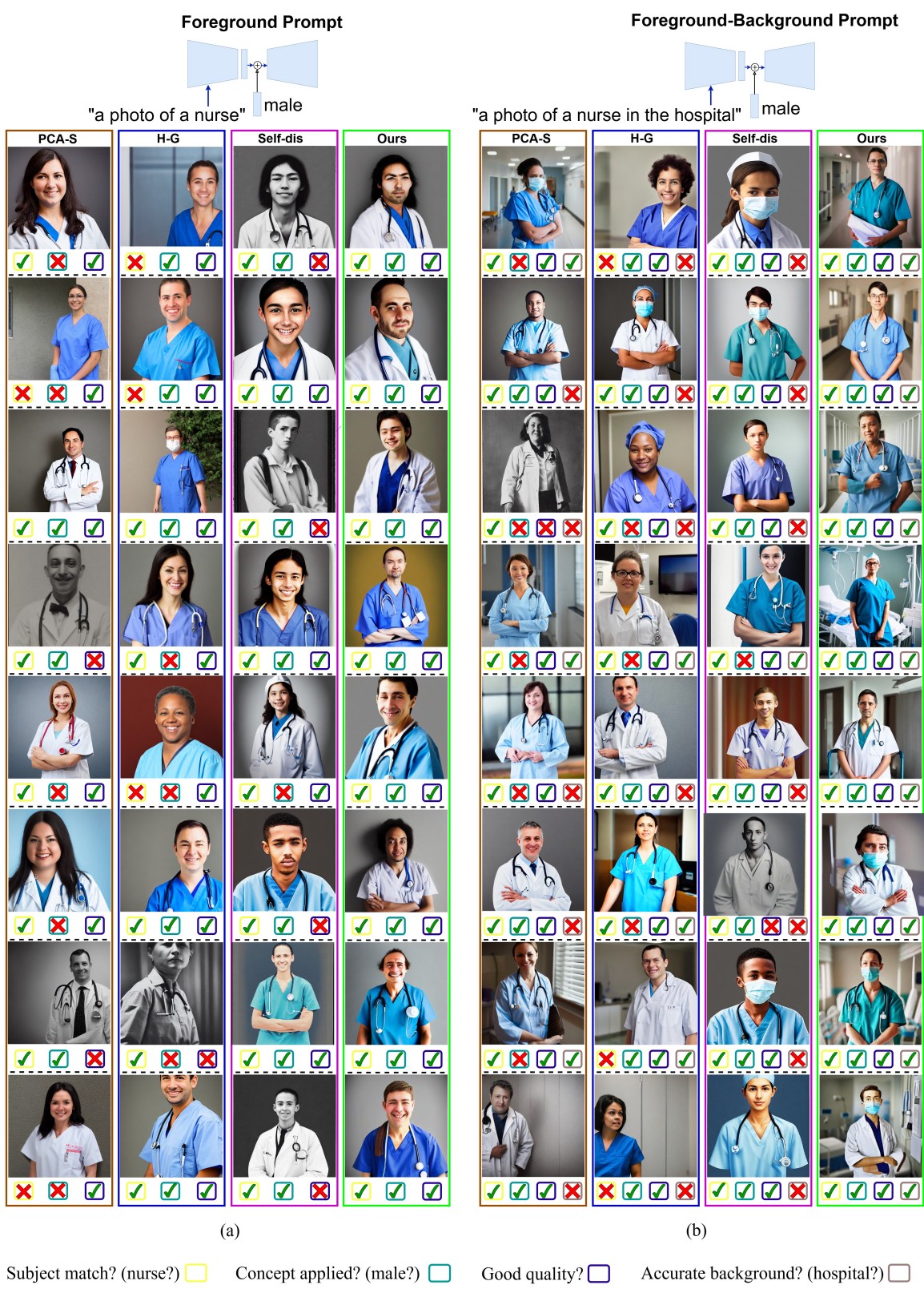

Figure 11: Comparison of image generation with the $\mathcal{T}$="male" concept across methods. (a) Foreground-only prompts: Our method effectively incorporates the male concept while improving image quality and maintaining subject identity. (b) Foreground-background prompts: Our approach produces realistic backgrounds while accurately integrating the concept and preserving prompt alignment.

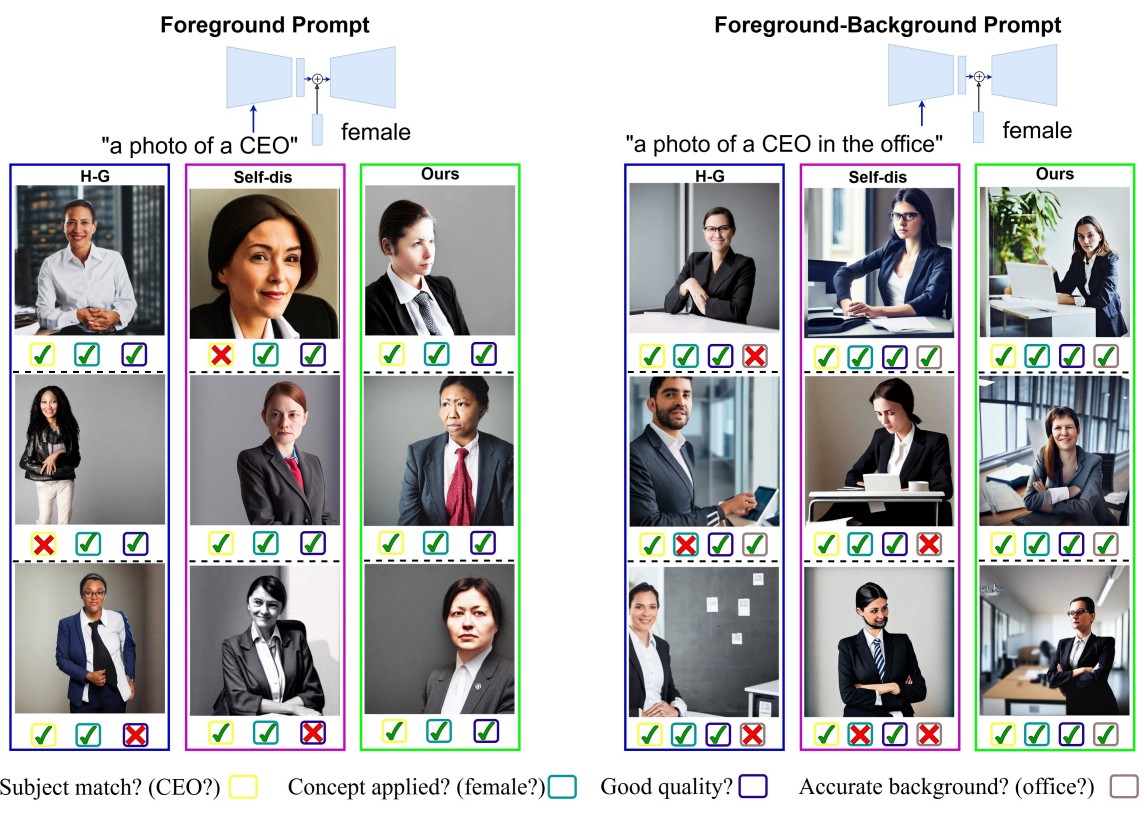

Figure 12: Comparison of image generation with the $\mathcal{T}=$"female" concept across different methods. In case (a), where the prompt specifies only the foreground (e.g., "a photo of a CEO"), our method successfully introduces the female concept, yielding higher visual quality while preserving the subject's identity. In case (b), when the prompt contains both subject and background elements (e.g., "a photo of a CEO in the office"), the proposed approach not only maintains accurate integration of the target concept but also generates realistic and coherent backgrounds, demonstrating strong alignment with the complete prompt.

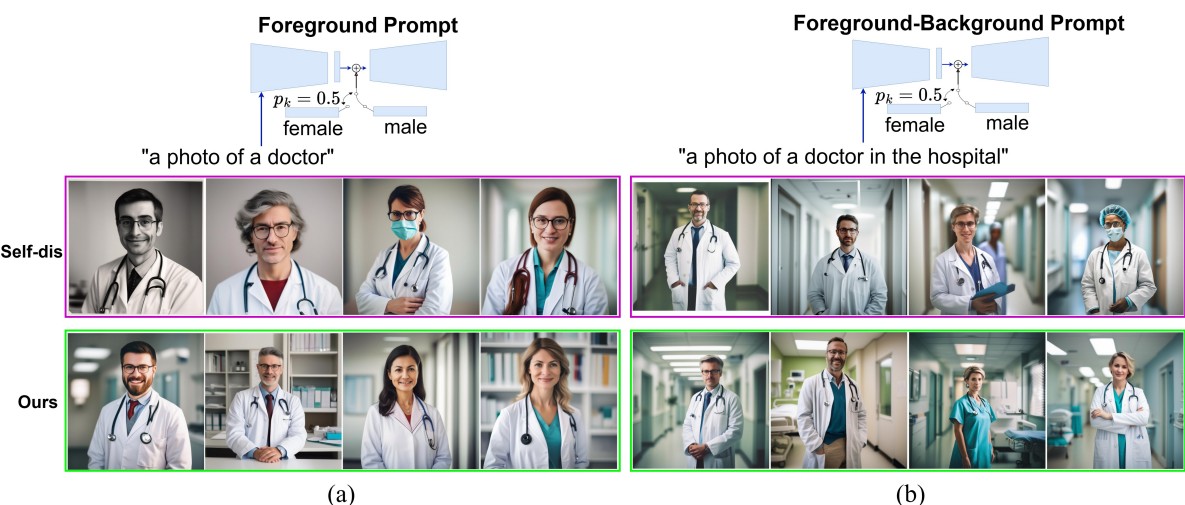

Figure 13: Comparison of unbiased image generation using the pre-trained SDXL model. (a) Foreground-only prompts: the proposed method enhances image quality, preserves subject identity, and improves fairness. (b) Foreground–background prompts: the method produces coherent backgrounds while maintaining fairness and strong prompt alignment.

| Heatmap Perturbation | $\Delta \downarrow$ | FID $\downarrow$ | CLIP$_f \uparrow$ |
|---|---|---|---|
| None (DAAM, default) | 0.12 | 0.66 | 0.36 |
| Jitter + Noise | 0.12 | 0.70 | 0.34 |

Table 14: Robustness to heatmap mislocalization and uncertainty. We perturb the attribution heatmap $\hat{\mathcal{I}}$ via spatial jitter and Gaussian noise, and report fairness ($\Delta$), image quality (FID), and semantic alignment (CLIP). Only modest degradation is observed under moderate perturbations, indicating that the proposed heatmap-guided masking and weighting mechanism is not brittle to heatmap noise.

# I  Ablation Study

## I.1  Robustness Analysis with Respect to Generated Heatmaps

In this section we explore the sensitivity of our method to heatmap quality. To that end, we performed a controlled mislocalization and noise stress test by perturbing the same DAAM heatmaps used in our pipeline before constructing the spatial weighting map $\mathbf{m}$ and the heatmap-guided loss weights $W = \mathbf{1} + \beta\hat{\mathcal{I}}$. Specifically, we apply (i) spatial jitter (translations of $\hat{\mathcal{I}}$ by 8 pixels) and (ii) Gaussian noise.

Our experimental setup evaluates robustness by learning the "female" and "male" concepts using perturbed heatmaps generated through spatial jitter (8 pixels) and Gaussian noise ($\sigma = 2$). Using both the perturbed concept vectors and the standard (unperturbed) ones, we generate 10 images for each of the 36 subjects. The quantitative results are reported in Table 14. Across these perturbations, our method exhibits only modest variations in FID and CLIP alignment metrics, which remain negligible. These results indicate that the proposed masking and weighting mechanism is robust and not sensitive to moderate heatmap mislocalization or uncertainty.

## I.2  Disentanglement Induced by Concept Suppression

To quantitatively evaluate disentangled attribute learning and address potential attribute leakage, we introduce a controlled intervention protocol that measures semantic changes induced by learned concept vectors. Using the same prompt ("a photo of a doctor") and identical random seeds, images are generated under three conditions. Let $\mathcal{I}_o$ denote the image produced by the original (unmodified) diffusion model, which serves as the baseline reference. Let $\mathcal{I}_{w/o}$ denote the image generated using a concept vector learned without spatial suppression, i.e., without applying the attribute-separation mask $\chi$ and spatial weighting map $\mathbf{m}$. Let $\mathcal{I}_w$ denote the image generated using a concept vector learned with the proposed heatmap-guided spatial learning framework, where both suppression mechanisms are applied. Keeping prompts and seeds fixed ensures that observed differences arise solely from the learned concept representations.

We focus on the target attribute $\mathcal{T} =$ "female", which the concept vector is intended to encode, and consider "old" as a spurious attribute that should not be introduced. Semantic alignment is quantified using CLIP similarity between attribute text prompts and generated images. The strength of target concept injection is measured as the change in CLIP similarity relative to the baseline image:

$$\delta_{\text{female}}^{(w/o)} = \left| \text{CLIP}(\text{"female"}, \mathcal{I}_{w/o}) - \text{CLIP}(\text{"female"}, \mathcal{I}_o) \right|, \tag{16}$$

$$\delta_{\text{female}}^{(w)} = \left| \text{CLIP}(\text{"female"}, \mathcal{I}_w) - \text{CLIP}(\text{"female"}, \mathcal{I}_o) \right|, \tag{17}$$

which capture the increase in semantic alignment with the intended attribute when learning concepts without suppression and with the proposed spatial framework, respectively.

To evaluate unintended semantic coupling, we similarly compute the change in CLIP similarity with respect to a spurious attribute:

$$\delta_{\text{old}}^{(w/o)} = \left| \text{CLIP}(\text{"old"}, \mathcal{I}_{w/o}) - \text{CLIP}(\text{"old"}, \mathcal{I}_o) \right|, \tag{18}$$

$$\delta_{\text{old}}^{(w)} = \left| \text{CLIP}(\text{"old"}, \mathcal{I}_w) - \text{CLIP}(\text{"old"}, \mathcal{I}_o) \right|. \tag{19}$$

| Method | $\delta_{\text{female}} \uparrow$ | $\delta_{\text{old}} \downarrow$ |
|---|---|---|
| Without applying $\mathbf{m}$ and $\chi$ | 0.07 | 0.08 |
| Proposed Heatmap-Guided Spatial Method (Ours) | **0.11** | **0.02** |

Table 15: Target attribute strength and spurious attribute leakage measured using CLIP similarity. Higher $\delta_{\text{female}}$ indicates stronger target concept injection, while lower $\delta_{\text{old}}$ indicates reduced spurious attribute leakage.

Higher $\delta_{\text{female}}$ indicates stronger encoding of the target concept, while lower $\delta_{\text{old}}$ reflects reduced spurious attribute leakage. Averaging these quantities over 200 independently generated samples allows us to directly assess whether suppressing attribute traces in the encoder and h-space leads to more exclusive concept encoding. The resulting quantitative comparison is reported in Table 15. The results show that our method increases target attribute alignment while substantially reducing spurious attribute shifts, providing quantitative evidence that the learned concept vectors capture the intended attribute more selectively.

### I.3   Ablation on the Choice of Manipulation Layer

We provide additional ablation experiments to justify the selection of the U-Net bottleneck representation (h-space) as the primary manipulation point for learning external concept vectors. To examine whether similar controllability can be achieved in other parts of the network, we introduce the trainable concept vector into the last encoder layer instead of the bottleneck. Although this intervention results in a partial reduction in demographic bias compared to the original pretrained diffusion model, the overall performance remains inferior to the proposed bottleneck manipulation. Fig. 14 presents a representative comparison for the prompt "a photo of a doctor in the hospital" where encoder-layer manipulation reduces bias relative to the original SD model but produces lower visual fidelity and less reliable attribute alignment than $h$-space control.

We further investigate the role of skip connections, which allow information to bypass the bottleneck. During concept learning, skip features are stored from trajectories corresponding to images containing the target attribute. At inference time, these stored features are injected under a neutral prompt without applying the learned concept vector in the bottleneck. The resulting generations still exhibit weak traces of the target attribute, indicating that skip pathways can carry attribute-related signals originating from training images even when the textual prompt does not explicitly contain the attribute. However, direct manipulation of skip features leads to less stable controllability and degraded generation quality compared to bottleneck intervention, as these representations do not demonstrate the same approximately linear semantic structure observed in h-space. Overall, these findings support the bottleneck as the most effective locus for learning disentangled and controllable concept directions while preserving generation fidelity. Comparison of concept-vector manipulation for the prompt "a photo of a doctor in the hospital", where the addition point of learnable concept vector is last layer of the encoder. While the bias is negligibility reduced compared to the original model, the performance remains dramatically inferior relative to the $h$-space approach.

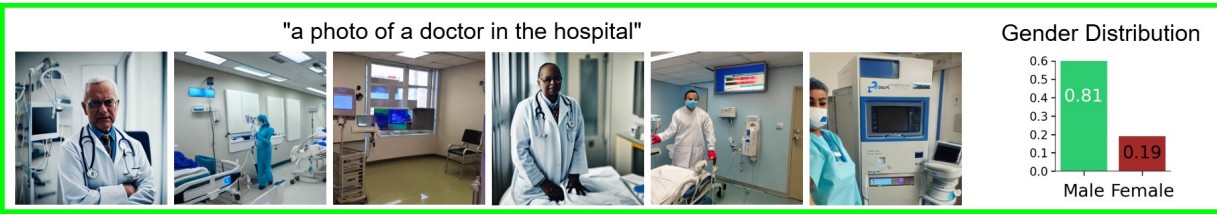

Figure 14: Ablation results for manipulation applied at the encoder layer (not $h$-space) on the prompts "a photo of a doctor in the hospital". While encoder-layer intervention negligibility reduces bias relative to the original SD model, it produces lower image fidelity and dramatically weaker attribute controllability compared to bottleneck-based manipulation.

### I.4   Ablation Study in Inference

To evaluate the impact of our proposed inference-time technique, we conduct an ablation study by comparing results with and without our method (see Eq. 5 and Fig. 3) on the prompts "a photo of a doctor in the hospital" and "a photo of a nurse in the hospital" in both SD and SDXL models. As illustrated in Fig. 15, incorporating the inference-time technique, shown in the bottom rows of Fig. 15(a) and Fig. 15(b), consistently enhances photorealism and background coherence in both models. Moreover, Fig. 15 presents quantitative metrics averaged over 36 subjects, where $\text{CLIP}_b$ scores are consistently higher with our method, demonstrating improved alignment between the generated images and the intended background terms.

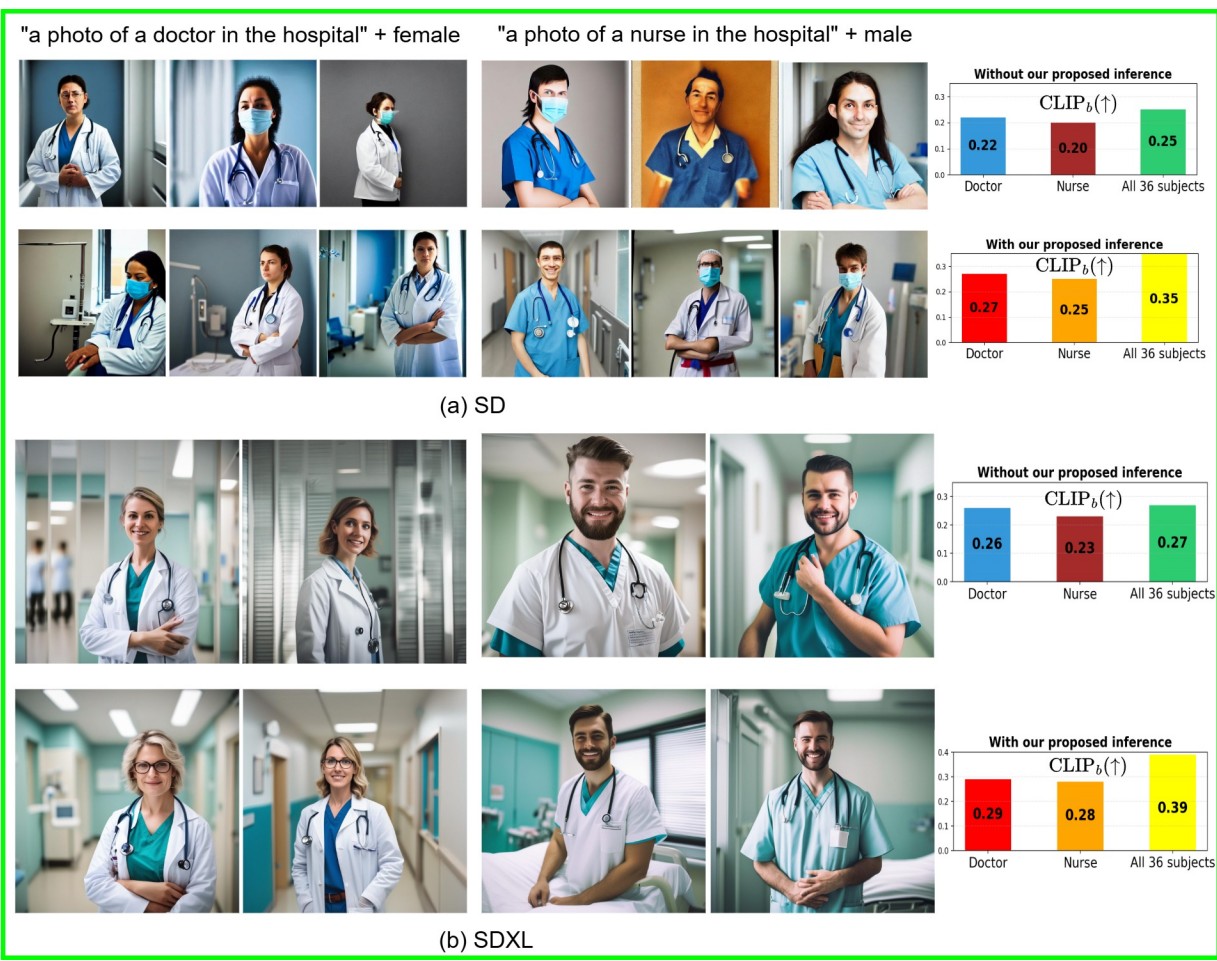

Figure 15: Ablation results for our inference method on the prompts "a photo of a doctor in the hospital" (with added concept $\mathcal{T}$ ="female") and "a photo of a nurse in the hospital" (with added concept $\mathcal{T}$ ="male"). Subfigure (a) shows results for SD and (b) for SDXL. In each case, the top rows depict generations without our inference method, while the bottom rows incorporate our proposed inference-time technique (i.e., applying Eq. 5). Visual comparisons highlight improved subject consistency and enhanced photorealism of hospital backgrounds. The accompanying bar charts further demonstrate that integrating our inference technique consistently increases the average $\text{CLIP}_b$ score across all 36 subjects.

## J   Extending Results for Diverse Prompt and Demographic Attributes

To further evaluate fairness beyond occupation-centric prompts, we construct an additional diverse prompts generated by ChatGPT 5.3, inspired by the compositional evaluation philosophy of CEB Wang et al. (2025). The new prompt set covers diverse real-world scenarios involving simultaneous demographic attributes,

specifically gender and age (e.g., young vs. old). For each prompt, we generate 100 images using identical sampling settings to ensure statistically reliable comparison across methods. Fairness performance is quantified using the deviation ratio metric, and the aggregated results are reported in Table 16. The proposed method consistently demonstrates improved balance across both gender and age groups under these broader prompting conditions. These findings provide additional empirical evidence that the fairness gains achieved by our approach generalize beyond occupation-focused benchmarks such as WinoBias, supporting its robustness under more diverse and compositionally rich evaluation settings.

To provide qualitative (subjective) insight, Fig. 16 presents example generations for the first prompt in Table 16, demonstrating balanced outcomes with respect to both gender and age under our proposed method. The results indicate that our approach effectively generates fair images while preserving background consistency and overall visual quality.

| Prompts | SDXL | | | |
| | Gender | | Age | |
| | Self-dis | **Ours** | Self-dis | **Ours** |
|---|---|---|---|---|
| a photo of a brain surgeon preparing for a complex operation in a modern hospital | 0.18 | 0.08 | 0.10 | 0.05 |
| a photo of a technology startup founder presenting a new product on stage | 0.11 | 0.05 | 0.08 | 0.03 |
| a cinematic shot of a senior executive leading a tense boardroom meeting | 0.21 | 0.14 | 0.26 | 0.16 |
| a documentary-style image of a renowned physicist giving a public lecture | 0.18 | 0.12 | 0.16 | 0.12 |
| a construction site worker operating heavy machinery at sunrise | 0.51 | 0.36 | 0.32 | 0.23 |
| a mechanic repairing a high-performance race car in a professional garage | 0.26 | 0.15 | 0.24 | 0.11 |
| a hospital caregiver comforting a patient in a quiet recovery room | 0.23 | 0.11 | 0.14 | 0.06 |
| a primary school classroom teacher guiding students during a reading lesson | 0.16 | 0.07 | 0.21 | 0.13 |
| a software developer working late at night in a small apartment workspace | 0.19 | 0.09 | 0.22 | 0.14 |
| a distinguished professor writing notes in a historic university office | 0.16 | 0.08 | 0.25 | 0.12 |
| Average | 0.22 | **0.12** | 0.20 | **0.11** |

Table 16: Fairness evaluation on demographically complex and broad prompts that implicitly induce gender and age bias. Results are measured using the deviation ratio $\Delta$ ($\downarrow$) on SDXL.

"a photo of a brain surgeon preparing for a complex operation in a modern hospital"

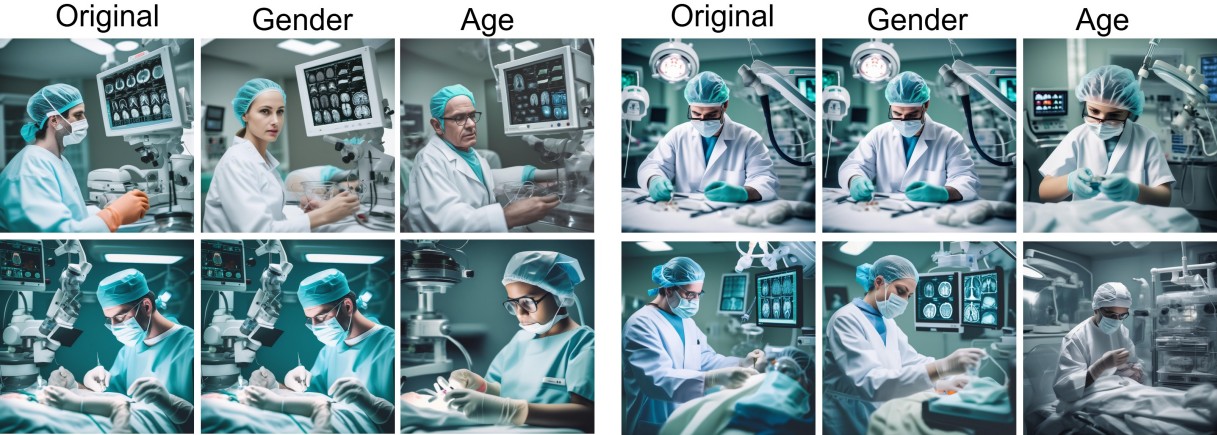

Figure 16: Qualitative examples for the first prompt in Table 16, showing balanced generation across gender and age. Our method maintains fairness while preserving background consistency and visual quality.

# K    Extending Results Beyond Human-Centric Attributes with Composability Property

To further examine the stability and composability of the learned concept vectors, we conduct an additional analysis evaluating prompt robustness across diverse conditions. In this experiment, a fixed learned concept vector and identical random seeds are applied to a range of human-centric and non-human-centric complex prompts. Representative results are shown in Fig. 17. The left side of the figure presents non-human-centric prompts, while the right side shows human-centric prompts. Note that the same learned concept vectors are applied in both cases. The observations indicate that the learned concept vectors function as reusable semantic controls that generalize consistently across prompt variations, rather than behaving as prompt-specific artifacts. These findings provide further evidence that the proposed manipulation captures a stable and interpretable semantic direction in the latent space. Furthermore, Fig. 18 illustrates the composability of multiple concepts across both human-centric and non-human-centric prompts, highlighting the effectiveness of linear combinations of different learned concept vectors in our method.

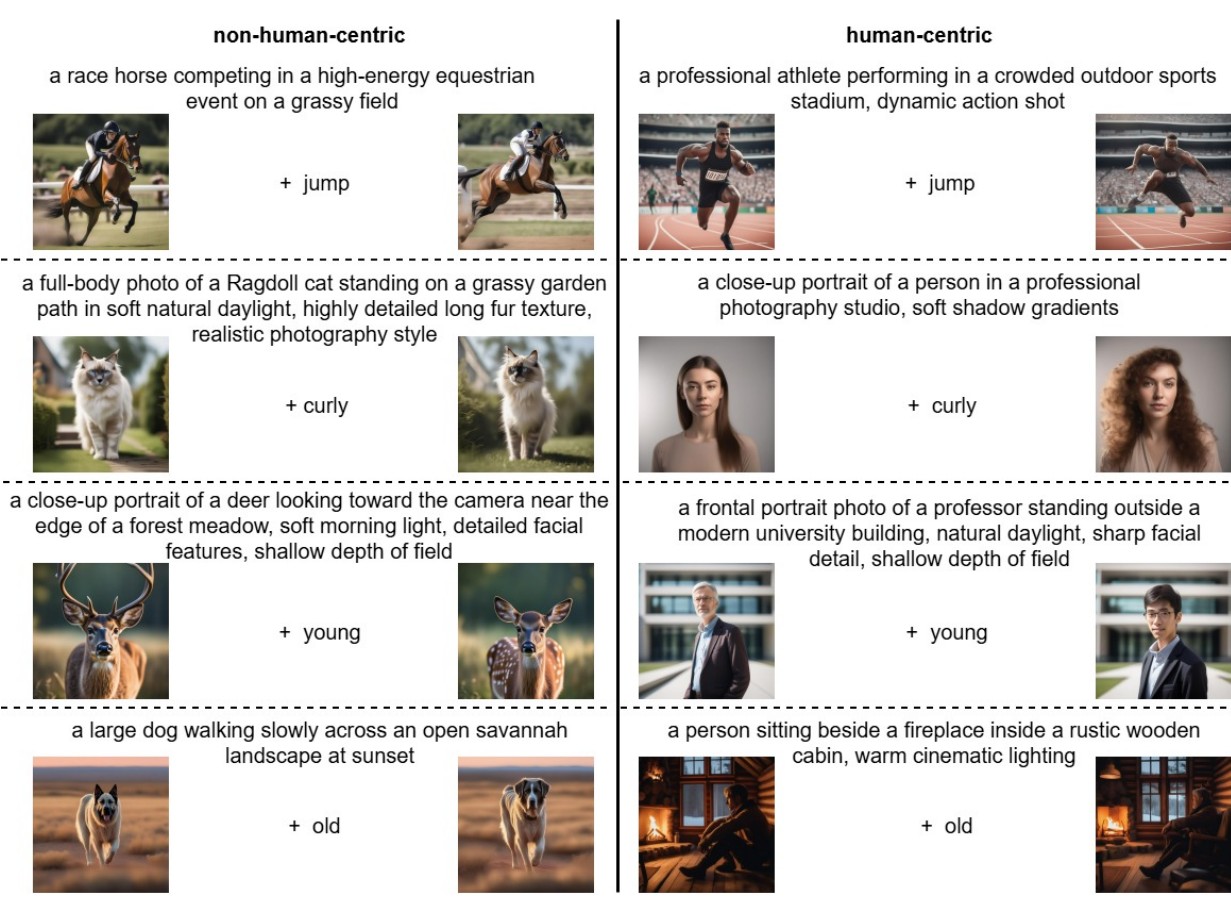

Figure 17: Stability of the learned concept vectors across diverse prompt conditions. The left side shows results for non-human-centric prompts, while the right side shows results for human-centric prompts. In all cases, the same learned concept vectors (jump, curly, young, and old) and identical random seeds are used. The results indicate that the intended attributes remain consistently observable across different scenarios, demonstrating that the learned concept vectors generalize reliably and function as stable semantic controls rather than prompt-specific manipulations.

a dog in a park in a bright sunny day

a professional athlete performing in a crowded outdoor sports stadium, dynamic action shot

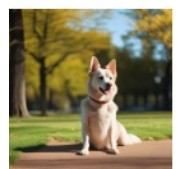

+ curly
+ jump
+ old

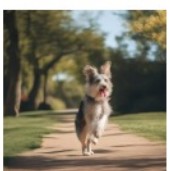

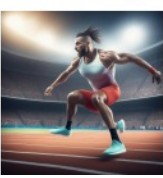

+ curly
+ jump
+ female

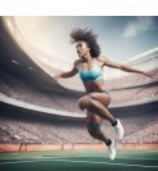

Figure 18: Composability of learned concept vectors across human-centric and non-human-centric prompts. Linear combinations of multiple concepts enable coherent attribute control while preserving the original scene semantics.

