# OpenReview forum: "Learning Where It Matters: Responsible and Interpretable Text-to-Image Generation with Background Consistency"
_TMLR — Accepted by TMLR_

### Review · Reviewer_XPEQ · 2026-02-24

**Summary Of Contributions:**

This paper studies the problem of interpretable concept control in text-to-image diffusion models through interventions in the U-Net bottleneck representation. The authors argue that existing h-space concept-learning methods rely on whole-image supervision, which can entangle the target attribute with spurious visual cues, leading to unintended bias and degraded background generation. To address this, the paper proposes a spatially focused learning framework that combines attribute-separation masking in cross-attention, bottleneck feature suppression via inverted attribution heatmaps, and a spatially weighted denoising objective to learn concept vectors. Also, the work introduces an inference-time enhancement mechanism that injects low-frequency components of the bottleneck representation to improve foreground–background consistency without retraining the model. The numerical analysis evaluates fairness and prompt alignment across Stable Diffusion backbones and presents ablations isolating the role of each component.

**Audience:**

Yes

**Audience Explanation:**

The paper addresses questions that are relevant to research on responsible generative modeling, especially in the context of post-hoc interventions in diffusion models. The focus on spatially localized concept learning in the bottleneck representation connects to the literature on interpretability and fairness. Therefore, I find the problem formulation and proposed mechanisms to be of interest to researchers working on representation effects and biases in generative models.

**Broader Impact Concerns:**

There is concern about the methodological and evaluation reliance on automated attribution and detection tools. The proposed learning pipeline depends on attribution heatmaps to spatially localize supervision, and the reported fairness and safety gains are measured using automated classifiers and content detectors. These tools are themselves imperfect and can suffer from systematic biases across demographics and visual contexts. Therefore, improvements obtained under these proxies may reflect alignment with the behavior of the attribution or detection models rather than genuine mitigation of bias or harmful content. This dependency should be stated more explicitly.

**Claims And Evidence:**

No

**Claims Explanation:**

The submission proposes an interesting framework and reports improvements using several quantitative metrics, but the evidence does not completely support some of the claims. Specifically, the claim that the learned concept vectors exclusively capture the target attribute is not directly validated. The evaluation relies on downstream proxy measures, such as deviation ratios and CLIP-based alignment, which are inadequate for establishing disentanglement or attribute specificity.

Also, the method depends critically on attribution heatmaps to spatially localize supervision, but the robustness of the approach to the choice and quality of the attribution signal is not examined. Since this component is the key to the formulation, the lack of sensitivity analysis weakens the empirical support. Furthermore, background consistency is evaluated using CLIP alignment, and the FID protocol measures divergence from the base model rather than fidelity to real data.

**Requested Changes:**

1- The claims regarding exclusive or disentangled attribute capture should be moderated or supported with more quantitative evidence. For example, the paper could include controlled interventions that measure attribute leakage, or evaluate whether applying the learned vector induces unintended shifts in other protected or semantic attributes.

2- The reliance on attribution heatmaps for spatial localization needs a robustness analysis. Since the masking and weighting mechanisms are constructed from these maps, the paper should test sensitivity to the choice of attribution method and to noise or mislocalization in the heatmaps.

3- The evaluation of background consistency should be done beyond CLIP-based alignment. Including at least one additional structural or semantic metric (e.g., object detection or segmentation consistency) would provide stronger evidence of improved foreground–background coherence.

4- The FID evaluation protocol should be clarified. Since the reference set consists of images generated by the base diffusion model rather than real data, the reported scores reflect distributional drift from the base model, not standard fidelity to real images. This distinction should be stated explicitly and the results interpreted accordingly.

---

> ### Author Response · Authors · 2026-03-20
>
> We sincerely thank the reviewer for the valuable comments and constructive suggestions. We have revised the manuscript accordingly and uploaded an updated version, with all changes highlighted in yellow.
>
> **Comment 1:** The claims regarding exclusive or disentangled attribute capture should be moderated or supported with more quantitative evidence. For example, the paper could include controlled interventions that measure attribute leakage, or evaluate whether applying the learned vector induces unintended shifts in other protected or semantic attributes.
>
> **Response 1:** Thank you for the thoughtful comment. In response, we added a new controlled evaluation explicitly aligned with our proposed method, where our method learns concept vectors by suppressing target-attribute features in the h-space using attribute-separation masks and attribute-attentive heatmaps prior to concept insertion (**added as Appendix I.2**).
> To quantitatively evaluate disentangled attribute learning and address potential attribute leakage, we introduce a controlled intervention protocol that measures semantic changes induced by learned concept vectors. Using the same prompt "a photo of a doctor" and identical random seeds, images are generated under three conditions. Let $\mathcal{I_o}$ denote the image produced by the original (unmodified) diffusion model, which serves as the baseline reference. Let $\mathcal{I_{w/o}}$ denote the image generated using a concept vector learned without spatial suppression, i.e., without applying the attribute-separation mask $\chi$ and spatial weighting map $\mathbf{m}$. Let $\mathcal{I}_{w}$ denote the image generated using a concept vector learned with the proposed heatmap-guided spatial learning framework, where both suppression mechanisms are applied. Keeping prompts and seeds fixed ensures that observed differences arise solely from the learned concept representations.
> We focus on the target attribute $\mathcal{T}=$"female" which the concept vector is intended to encode and consider "old" as a spurious attribute that should not be introduced. Semantic alignment is quantified using CLIP similarity between attribute text prompts and generated images. The strength of target concept injection is measured as the change in CLIP similarity relative to the baseline image:
>
> $\delta_{\mathrm{female}}^{(w/o)} = \left| \mathrm{CLIP}(\text{female}, \mathcal{I_{w/o}}) - \mathrm{CLIP}(\text{female}, \mathcal{I_o}) \right|$,
>
> $\delta_{\mathrm{female}}^{(w)} = \left| \mathrm{CLIP}(\text{female}, \mathcal{I}_w) - \mathrm{CLIP}(\text{female}, \mathcal{I_o}) \right|$,
>
> which capture the increase in semantic alignment with the intended attribute when learning concepts without suppression and with the proposed spatial framework, respectively.
>
> To evaluate unintended semantic coupling, we similarly compute the change in CLIP similarity with respect to a spurious attribute:
>
> $\delta_{\text{old}}^{(w/o)}=\left|\mathrm{CLIP}(\text{old}, \mathcal{I_{w/o}})-\mathrm{CLIP}(\text{old}, \mathcal{I_o})\right|$,
>
> $\delta_{\text{old}}^{(w)}=\left|\mathrm{CLIP}(\text{old}, \mathcal{I_w})-\mathrm{CLIP}(\text{old}, \mathcal{I_o})\right|.$
>
> Higher $\delta_{\text{female}}$ indicates stronger encoding of the target concept, while lower $\delta_{\text{old}}$ reflects reduced spurious attribute leakage. Averaging these quantities over 200 independently generated samples allows us to directly assess whether suppressing attribute traces in the encoder and h-space leads to more exclusive concept encoding. The resulting quantitative comparison is reported in **Table 15 in Appendix I.2**. The results show that our method increases target attribute alignment while substantially reducing spurious attribute shifts, providing quantitative evidence that the learned concept vectors capture the intended attribute more selectively.
>
> | Method |  $\delta_{\text{female}}$ ↑ | $\delta_{\text{old}}$ ↓ |
> |--------|------------|---------|
> | Without applying $\mathbf{m}$ and $\chi$ | 0.07 | 0.08 |
> | Proposed Heatmap-Guided Spatial Method (Ours) | **0.11** | **0.02** |
>
> **Table 15:** Target attribute strength and spurious attribute leakage measured using CLIP similarity. Higher $\delta_{\text{female}}$ indicates stronger target concept injection, while lower $\delta_{\text{old}}$ indicates reduced spurious attribute leakage.

---

> ### Author Response · Authors · 2026-03-20
>
> **Comment 2:** The reliance on attribution heatmaps for spatial localization needs a robustness analysis. Since the masking and weighting mechanisms are constructed from these maps, the paper should test sensitivity to the choice of attribution method and to noise or mislocalization in the heatmaps.
>
> **Response 2:** Thank you for the thoughtful comment. We agree that reliance on attribution heatmaps warrants robustness analysis. To minimize confounds and isolate sensitivity to heatmap quality, we performed a controlled mislocalization and noise stress test by perturbing the same DAAM heatmaps used in our pipeline before constructing the spatial weighting map $\mathbf{m}$ and the heatmap-guided loss weights $W = \mathbf{1} + \beta \hat{\mathcal{I}}$. Specifically, we apply (i) spatial jitter (translations of $\hat{\mathcal{I}}$ by 8 pixels) and (ii) Gaussian noise, and we have added **Table 14 to Appendix I.1** of the revised manuscript to show this sensitivity.
>
> Our experimental setup evaluates robustness by learning the “female” and “male” concepts using perturbed heatmaps generated through spatial jitter (8 pixels) and Gaussian noise ($\sigma = 2$). Using both the perturbed concept vectors and the standard (unperturbed) ones, we generate 10 images for each of the 36 subjects. The quantitative results are reported in Table 14. Across these perturbations, our method exhibits only modest variations in FID and CLIP alignment metrics, which are negligible. These results indicate that the proposed masking and weighting mechanism is robust and not sensitive to moderate heatmap mislocalization or uncertainty.
>
> | Heatmap Perturbation | Δ ↓ | FID ↓ | CLIP$_f$ ↑ |
> |----------------------|-----|-------|----------|
> | None (DAAM, default) | 0.12 | 0.66 | 0.36 |
> | Jitter + Noise | 0.12 | 0.70 | 0.34 |
>
> **Table 14:** Robustness to heatmap mislocalization and uncertainty. We perturb the attribution heatmap $\hat{\mathcal{I}}$ via spatial jitter and Gaussian noise, and report fairness (Δ), image quality (FID), and semantic alignment (CLIP). Only modest degradation is observed under moderate perturbations, indicating that the proposed heatmap-guided masking and weighting mechanism is not brittle to heatmap noise.
>
>
> **Comment 3:** The evaluation of background consistency should be done beyond CLIP-based alignment. Including at least one additional structural or semantic metric (e.g., object detection or segmentation consistency) would provide stronger evidence of improved foreground–background coherence.
>
> **Response 3:** We thank the reviewer for the suggestion to evaluate background consistency beyond CLIP-based alignment. In response, we added an additional structure-aware evaluation based on background object recall using a pretrained GroundingDINO detector, reported in **the last row of Table 1 (Subsection 4.1)** in the revised manuscript. The following corresponding description has also been added to Section 4 (Metrics).
>
> In addition, to evaluate structural background consistency beyond semantic similarity, we measure background object preservation using a pretrained open-vocabulary object detector (GroundingDINO [Liu et al., 2023]). For each prompt-conditioned generation, objects detected in the image produced by the original Stable Diffusion (SD) model are treated as reference background semantics. The same detector is applied to the corresponding images generated by the proposed method and competing baselines. Background consistency is quantified using background object recall [Liu et al., 2023; Ghosh et al., 2023]. Let $O_o$ denote the set of background object categories detected in the original SD generation and $O_w$ denote the corresponding set detected in a method-generated image. Background consistency is defined as the proportion of reference background objects that remain detectable:
>
> Object Recall $= \| \( O_o \cap O_w \) \| / \| \( O_o \) \| $
>
> Higher object recall indicates stronger preservation of background scene composition relative to the prompt-conditioned reference generation from the original SD model (please see **the last row of Table 1 (Subsection 4.1)** in the revised manuscript).

---

> ### Author Response · Authors · 2026-03-20
>
> **Comment 4:** The FID evaluation protocol should be clarified. Since the reference set consists of images generated by the base diffusion model rather than real data, the reported scores reflect distributional drift from the base model, not standard fidelity to real images. This distinction should be stated explicitly and the results interpreted accordingly.
>
> **Response 4:** We thank the reviewer for pointing out the need to clarify the interpretation of the FID evaluation protocol. We agree that, in our setting, the reference distribution differs from the conventional use of FID against real-image datasets, and we have revised the manuscript to explicitly state this distinction. It is worth noting that we also evaluate our method on the real-image COCO-30k dataset, where FID is computed between generated images and real images.
>
> This protocol is intentionally adopted because our objective is not to retrain or replace the generative model, but to modify generation through h-space manipulation while preserving the visual characteristics and photorealism of the underlying model. Using the base model outputs as the reference allows us to measure whether the proposed method maintains generation quality relative to the original model behavior. Under this interpretation, lower FID indicates reduced unintended distributional drift and better preservation of the base model’s visual fidelity.
>
> To avoid ambiguity, we have added explicit clarification in **Sec. 4 (Metrics)** of the revised manuscript stating that:
>
> "Unlike standard FID evaluation against real-image datasets, we compute FID using images generated by the original diffusion model as the reference distribution (except for experiments conducted on the COCO-30k dataset, where real images are used). Consequently, this metric measures distributional deviation from the base model rather than absolute realism with respect to real-world image distributions."
>
> **Broader Impact Concerns:**
> There is concern about the methodological and evaluation reliance on automated attribution and detection tools. The proposed learning pipeline depends on attribution heatmaps to spatially localize supervision, and the reported fairness and safety gains are measured using automated classifiers and content detectors. These tools are themselves imperfect and can suffer from systematic biases across demographics and visual contexts. Therefore, improvements obtained under these proxies may reflect alignment with the behavior of the attribution or detection models rather than genuine mitigation of bias or harmful content. This dependency should be stated more explicitly.
>
> **Response:** We thank the reviewer for raising this important point. We agree that attribution methods and automated evaluation tools, including classifiers and content detectors, are imperfect and may themselves exhibit biases across demographic groups and visual contexts. However, such tools are widely adopted in the generative modeling and fairness literature as scalable evaluation proxies, and our evaluation protocol follows established community practices to ensure comparability and reproducibility.
>
> In our work, these tools are used as measurement proxies rather than as sources of supervision defining fairness objectives. Attribution heatmaps are used to spatially localize learning signals, while standardized automated evaluators enable consistent and reproducible comparisons across methods. Accordingly, our claims focus on relative improvements under a fixed evaluation protocol rather than absolute guarantees of bias mitigation or safety.
>
> We have revised the manuscript to explicitly acknowledge this dependency and clarify that reported improvements should be interpreted as alignment gains measured under commonly used evaluation proxies by adding the following explanation in **Section 4**:
>
> "While attribution methods and automated evaluation tools may have inherent limitations and potential biases, they remain widely adopted evaluation proxies in the generative modeling and fairness literature. Accordingly, the reported improvements should be interpreted as relative gains under a standardized and commonly used evaluation protocol."

---

### Review · Reviewer_rpuc · 2026-03-07

**Summary Of Contributions:**

This paper addresses bias and background incoherence in text-to-image diffusion models by proposing a dual-pronged approach to manipulate the U-Net bottleneck (h-space). The primary contributions include a spatially focused concept learning framework during training that disentangles target attributes using cross-attention suppression and a spatially weighted reconstruction loss, alongside an inference-time strategy that enhances low-frequency components to maintain background consistency. While the method's key strengths lie in its comprehensive, well-motivated design that successfully improves fairness, subject fidelity, and background coherence without sacrificing overall visual quality, it also has notable weaknesses. Specifically, its effectiveness relies heavily on the accuracy of cross-attention maps to localize attributes; it may introduce computational overhead, and its strict dependency on the U-Net architecture could limit its generalizability to newer transformer-based diffusion models.

**Audience:**

Yes

**Audience Explanation:**

Alignment in text-to-image (T2I) generation is a broad field that has garnered significant attention due to its rapid development and wide applicability. This paper aims to address key issues in concept alignment, fairness, and visual quality, which represent the core challenges in T2I generation.

**Broader Impact Concerns:**

While this work inherently focuses on positive ethical impacts—specifically improving fairness and safety in text-to-image models—it introduces a dual-use risk that warrants discussion in a Broader Impact Statement. By explicitly isolating and granting precise linear control over demographic and semantic "concept vectors" (e.g., race, gender) in the latent space, the methodology could theoretically be maliciously applied by bad actors to intentionally inject stereotypes, amplify biases, or generate targeted disinformation. The authors should explicitly acknowledge this dual-use potential and discuss potential safeguards or limitations regarding the subjective categorization of sensitive demographic attributes used during concept learning.

**Claims And Evidence:**

Yes

**Claims Explanation:**

The paper wants to address 3 issues in T2I generation: alignment, fairness, and visual quality. The authors do not rely on a single metric or model; rather, they evaluate fairness, text-alignment, and visual quality using standard, widely accepted metrics (FID, CLIP, Deviation Ratio). Furthermore, their demonstration that the method scales successfully from SD v1.4 to the much larger SDXL model establishes strong generalizability for their claims.

**Requested Changes:**

1. The proposed methodology introduces several additional operations compared to baseline h-space methods. Please provide a concrete quantitative analysis of the computational overhead against baseline methods.

2. Like the Self-dis method, this approach manipulates the h-space (i.e., the U-Net bottleneck). Have the authors considered or conducted an ablation study on the choice of this specific layer compared to other layers within the encoder or decoder? While targeting the bottleneck is intuitive for semantic manipulation, the presence of skip-connections in the U-Net architecture suggests that information bypasses the bottleneck, which may mean the difference between manipulating the h-space versus other layers is not as drastic as presumed. Exploring this would strengthen the justification for isolating the bottleneck.

3. The core of the spatially focused learning framework relies heavily on the accuracy of the cross-attention maps. Could the authors clarify exactly which cross-attention layers are being utilized to generate these maps? Furthermore, including an ablation study examining the impact of using different cross-attention layers (e.g., early vs. late encoder layers) would provide valuable insight into the method's robustness.

---

> ### Author Response · Authors · 2026-03-20
>
> We sincerely thank the reviewer for the valuable feedback and constructive suggestions. We have carefully revised the manuscript in response and uploaded an updated version, with all **modifications highlighted in yellow**.
>
> **Comment 1:** The proposed methodology introduces several additional operations compared to baseline h-space methods. Please provide a concrete quantitative analysis of the computational overhead against baseline methods.
>
> **Response 1:** We thank the reviewer for raising this important point regarding the computational overhead of the proposed methodology. To address this concern, we provide a quantitative runtime comparison with baseline h-space methods, separating the computation into three stages that correspond to Algorithm 1 (data generation), Algorithm 2 (learning), and Algorithm 3 (inference). A detailed explanation together with quantitative results has been added to **Appendix F**.
>
> Importantly, several components introduced by our method, specifically the generation of attribute-attentive heatmaps and the construction of attribute-separation masks, are executed during a dataset preparation stage prior to optimization. These quantities are computed once and stored, and therefore they do not introduce repeated overhead during the training iterations. As a result, their cost should be interpreted as one-time preprocessing overhead rather than recurring training cost.
>
> To make this distinction explicit, **Table 11 in Appendix F** reports the comparison of the running time for Algorithm 1 (data generation), Algorithm 2 (learning), and Algorithm 3 (inference). The data generation stage includes operations required to prepare the training dataset, while the training stage reflects the actual optimization procedure that is directly comparable across methods. All simulations were conducted on an NVIDIA H100 GPU with 80 GB of memory.
>
> During Algorithm 2, the additional computation introduced by our method is limited to an additional encoder-side pass associated with the heatmap input and lightweight element-wise spatial operations used in the proposed suppression and weighted loss formulations. Since the backbone diffusion model remains frozen, the resulting overhead in the training loop is modest relative to the baseline h-space optimization procedure. In Algorithm 3, the only additional computation introduced by our method is the application of a low-frequency enhancement step, which involves computing the Discrete Wavelet Transform (DWT) of the h-vector.
>
> | Stage | Metric (second) | H-G | Self-dis | Ours |
> |------|------------------|-----|----------|------|
> | Alg. 1: Data Generation | Single training image generation time | 0.96 | 0.96 | 1.48 |
> | Alg. 2: Learning | Time for single training epoch (1k images) | 78 | 47 | 71 |
> | Alg. 3: Inference | Inference time | 3.10 | 2.36 | 3.78 |
>
> **Table 11:** Runtime comparison for Algorithm 1 (data generation), Algorithm 2 (concept vector learning), and Algorithm 3 (inference) across different h-space methods using an NVIDIA H100 GPU with 80 GB memory.

---

> ### Author Response · Authors · 2026-03-20
>
> **Comment 2:** Like the Self-dis method, this approach manipulates the h-space (i.e., the U-Net bottleneck). Have the authors considered or conducted an ablation study on the choice of this specific layer compared to other layers within the encoder or decoder? While targeting the bottleneck is intuitive for semantic manipulation, the presence of skip-connections in the U-Net architecture suggests that information bypasses the bottleneck, which may mean the difference between manipulating the h-space versus other layers is not as drastic as presumed. Exploring this would strengthen the justification for isolating the bottleneck.
>
> **Response 2:** We appreciate the reviewer’s suggestion regarding the role of other layers in the U-Net. To address this concern, we have added an additional exploration in **Appendix I.3**, where the external trainable concept vector is introduced into the last encoder layer instead of the bottleneck h-space. However, these alternatives do not produce comparable results. Empirically, the bottleneck representation exhibits a more semantically structured and approximately linear behavior, which enables attribute directions to be effectively captured through simple additive concept vectors. In **Figure 14 (Appendix I.3)**, we present results for the prompt "a photo of a doctor in the hospital", where the learnable concept vector is injected at the last encoder layer. Although bias is slightly reduced compared to the original model, the overall performance remains significantly inferior to the h-space manipulation approach.
>
> To further examine the reviewer’s concern regarding skip connections, we conduct an intervention experiment in which skip features are first stored while learning the concept "female" under a neutral prompt (e.g., "a person"). During inference with the prompt "a photo of a doctor in the hospital", we replace the corresponding skip features with the stored ones, without introducing the learned concept vector at the bottleneck. The resulting generations still exhibit a weak trace of the target attribute, suggesting that skip pathways can carry attribute-related information. However, unlike the bottleneck h-space, skip connections do not demonstrate an approximately linear semantic behavior: additive manipulation at these locations does not reliably introduce the target attribute, and direct feature replacement yields weaker controllability together with reduced visual quality compared to bottleneck manipulation.
>
> Overall, this experiment indicates that skip connections may encode attribute-specific cues originating from the training images even when the text prompt does not explicitly contain the attribute (e.g., "a person"). These findings further support the design choice of isolating the bottleneck representation as the primary locus for learning controllable concept directions.
> (Please see **Figure 14 in Appendix I.3** of the revised manuscript).

---

> ### Author Response · Authors · 2026-03-20
>
> **Comment 3:** The core of the spatially focused learning framework relies heavily on the accuracy of the cross-attention maps. Could the authors clarify exactly which cross-attention layers are being utilized to generate these maps? Furthermore, including an ablation study examining the impact of using different cross-attention layers (e.g., early vs. late encoder layers) would provide valuable insight into the method's robustness.
>
> **Response 3:** Thank you for the thoughtful comment. We agree that reliance on attribution heatmaps warrants robustness analysis. To minimize confounds and isolate sensitivity to heatmap quality, we performed a controlled mislocalization and noise stress test by perturbing the same DAAM heatmaps used in our pipeline before constructing the spatial weighting map $\mathbf{m}$ and the heatmap-guided loss weights $W = \mathbf{1} + \beta \hat{\mathcal{I}}$. Specifically, we apply (i) spatial jitter (translations of $\hat{\mathcal{I}}$ by 8 pixels) and (ii) Gaussian noise, and we have added **Table 14 to Appendix I.1** of the revised manuscript to show this sensitivity.
>
> Our experimental setup evaluates robustness by learning the “female” and “male” concepts using perturbed heatmaps generated through spatial jitter (8 pixels) and Gaussian noise ($\sigma = 2$). Using both the perturbed concept vectors and the standard (unperturbed) ones, we generate 10 images for each of the 36 subjects. The quantitative results are reported in Table 14. Across these perturbations, our method exhibits only modest variations in FID and CLIP alignment metrics, which are negligible. These results indicate that the proposed masking and weighting mechanism is robust and not sensitive to moderate heatmap mislocalization or uncertainty.
>
> | Heatmap Perturbation | Δ ↓ | FID ↓ | CLIP$_f$ ↑ |
> |----------------------|-----|-------|----------|
> | None (DAAM, default) | 0.12 | 0.66 | 0.36 |
> | Jitter + Noise | 0.12 | 0.70 | 0.34 |
>
> **Table 14:** Robustness to heatmap mislocalization and uncertainty. We perturb the attribution heatmap $\hat{\mathcal{I}}$ via spatial jitter and Gaussian noise, and report fairness (Δ), image quality (FID), and semantic alignment (CLIP). Only modest degradation is observed under moderate perturbations, indicating that the proposed heatmap-guided masking and weighting mechanism is not brittle to heatmap noise.
>
> On the other hand, the attribute-separation mask $\chi$ used to suppress target-attribute features within the U-Net is derived from encoder cross-attention and applied at the final encoder MCA layer. This design is motivated by the fact that the final encoder layer directly precedes the bottleneck h-space where the concept vector is learned, making it a natural location for spatially targeted intervention.
>
> To further examine the effect of the layer choice, an ablation study analyzing the impact of using cross-attention across all encoder layers is provided in the bottom part of Table 9 in Subsection E.3. Although applying $\chi$ across all layers slightly reduces the deviation ratio, the improvement is negligible relative to the additional computational cost. Since the final encoder layer contributes most directly to the h-space representation, masking only this layer achieves nearly identical performance with substantially lower complexity.
>
> | Setting | Δ ↓ |
> |--------|-----|
> | One-level wavelet decomposition | **0.10** |
> | Two-level wavelet decomposition | 0.12 |
> | $\chi$ applied only at final layer (l = L) | 0.101 |
> | $\chi$ applied to all layers | **0.096** |
>
> **Table 9:** Ablation study on the number of wavelet decomposition levels and the placement of the attribute-separation mask $\chi$. Results are reported using the deviation ratio Δ (lower is better). One-level wavelet decomposition and applying $\chi$ only to the final layer provide the best trade-off between performance and computational efficiency.

---

> ### Author Response · Authors · 2026-03-20
>
> **Broader Impact Concerns:**
> While this work inherently focuses on positive ethical impacts—specifically improving fairness and safety in text-to-image models—it introduces a dual-use risk that warrants discussion in a Broader Impact Statement. By explicitly isolating and granting precise linear control over demographic and semantic “concept vectors” (e.g., race, gender) in the latent space, the methodology could theoretically be maliciously applied by bad actors to intentionally inject stereotypes, amplify biases, or generate targeted disinformation. The authors should explicitly acknowledge this dual-use potential and discuss potential safeguards or limitations regarding the subjective categorization of sensitive demographic attributes used during concept learning.
>
> **Response:** We thank the reviewer for raising this important point. We agree that methods enabling explicit control over semantic concept representations may introduce potential dual-use considerations. In response, we have added a discussion to **Section 6 (Impact Statement)** acknowledging that such mechanisms could theoretically be misused to inject or amplify biased attributes, as follows:
>
> "While our work aims to improve fairness and safety in text-to-image models, we acknowledge that explicit control over semantic concept vectors could introduce dual-use considerations. In principle, such mechanisms could be misused to intentionally inject or amplify biased attributes. However, similar controllability already exists in many generative modeling approaches. Our goal is to provide a transparent and interpretable framework for identifying, auditing, and mitigating attribute leakage and bias in diffusion models rather than leaving such associations implicitly embedded in opaque latent representations."

---

### Review · Reviewer_iTg4 · 2026-03-11

**Summary Of Contributions:**

This paper tackles the problem of responsible and controllable text-to-image generation, where T2I diffusion models often produce biased outputs, misaligned subjects, and inconsistent backgrounds when manipulating semantic attributes. To address this, the authors propose a spatially focused concept learning framework in the U-Net bottleneck (h-space). Specifically, it learns disentangled concept vectors by suppressing attribute-related signals in the base model using an attribute-separation mask in cross-attention, a spatial weighting mask applied to the bottleneck features, and a spatially weighted reconstruction loss that focuses learning on relevant regions. In addition, the authors introduce a plug-and-play inference-time method that improves foreground–background consistency by enhancing low-frequency components of the h-space representation using a wavelet transform. Experiments on SD v1.4 and SDXL demonstrate improved fairness, image quality, and prompt alignment compared to prior h-space methods (e.g., PCA-S, H-G, Self-dis), with lower bias deviation ratios, improved FID scores, and higher CLIP-based alignment metrics on benchmarks such as WinoBias and COCO-30k, while also improving safety performance on the I2P dataset.

**Audience:**

Yes

**Audience Explanation:**

The paper addresses responsible and controllable text-to-image generation in diffusion models, which is an important, yet overlooked research area within generative modeling, interpretability, and AI safety. In particular, the work focuses on manipulating the U-Net bottleneck latent space (h-space) to obtain interpretable and controllable semantic directions, a topic that has been studied throughout the diffusion model research community. The proposed approach for spatially disentangling attribute representations and improving background consistency during inference would likely be of interest to researchers working on diffusion model controllability, fairness in generative models, and interpretable latent representations.

That said, the paper’s impact may be somewhat narrower because it focuses specifically on h-space manipulation in U-Net–based diffusion models, which is bit outdated. As such, if the authors could verify the claims on other diffusion models (e.g., diffusion transformer based ones), it would attract more audience in the community.

**Broader Impact Concerns:**

The work aims to improve fairness and safety in text-to-image models, which has positive societal implications. However, the ability to manipulate semantic attributes via concept vectors could potentially be misused to intentionally generate biased or misleading images. In addition, the fairness evaluation is limited to specific datasets and attributes, leaving open questions about broader demographic coverage. A brief discussion of potential misuse and limitations of the fairness evaluation would strengthen the broader impact considerations.

**Claims And Evidence:**

Yes

**Claims Explanation:**

The proposed claims are *partially* supported by evidences.

The paper provides several empirical results demonstrating improvements over prior h-space methods, including quantitative evaluations on WinoBias, COCO-30k, and I2P benchmarks using metrics such as deviation ratio (fairness), FID (image quality), and CLIP-based alignment scores. The reported results show consistent gains in fairness, prompt alignment, and background consistency compared to baselines such as PCA-S, H-G, and Self-dis, and the paper also includes ablation studies analyzing the contributions of the spatial masking components and the inference-time low-frequency enhancement. These experiments provide evidence that the proposed mechanisms can improve attribute disentanglement and background generation in U-Net–based diffusion models.

However, the evidence is not fully convincing in several aspects. First, the experiments are restricted to U-Net architectures (SD v1.4 and SDXL), leaving the generality of the method to other diffusion models (e.g., transformer based, which is widely used in practice) unclear. Second, some of the evaluation protocols rely on generated reference images or proxy metrics (e.g., CLIP alignment), which may not fully capture improvements in semantic correctness or fairness. Third, while qualitative examples suggest improved background consistency, the causal role of the proposed low-frequency enhancement mechanism is only indirectly validated.

Overall, while the presented experiments support the paper’s claims to a reasonable extent, additional validation across more architectures, prompts, and datasets would strengthen the evidence.

**Requested Changes:**

1. While the paper includes ablation studies for the spatial masks and the weighted loss, the causal contribution of each component (e.g., the low-frequency enhancement during inference) remains somewhat indirect. Providing more detailed analysis or additional controlled experiments would strengthen the claim that it improves background consistency rather than generally altering image statistics.

2. The fairness evaluation primarily relies on the WinoBias benchmark and deviation ratio metric, which may not fully capture biases in broader real-world prompts. Including additional datasets or evaluation protocols (e.g., more diverse prompts or demographic attributes) would make the fairness claims more convincing.

3. The experiments are limited to U-Net–based diffusion models which are outdated. Since the field is increasingly moving toward transformer-based diffusion models, the authors should discuss limitations more explicitly or provide preliminary experiments demonstrating how the method might extend beyond U-Net architectures.

4. The concept vector learning procedure involves multiple components (attribute-separation masks, spatial weighting maps, and weighted loss). A more concise algorithmic description summarizing the method would improve readability.

5. The method introduces additional operations (attention masking, heatmap computation, wavelet transforms). Reporting training and inference overhead compared to baseline h-space methods would be helpful to understand practical trade-offs.

6. The concept vector manipulation experiments are interesting; further discussion or analysis on how stable and composable these vectors are across prompts and seeds would strengthen the interpretability claims.

---

> ### Author Response · Authors · 2026-03-20
>
> We sincerely appreciate the reviewer’s insightful comments and constructive suggestions. The manuscript has been revised accordingly, and an updated version with all changes **highlighted in yellow** has been uploaded.
>
> **Comment 1:** While the paper includes ablation studies for the spatial masks and the weighted loss, the causal contribution of each component (e.g., the low-frequency enhancement during inference) remains somewhat indirect. Providing more detailed analysis or additional controlled experiments would strengthen the claim that it improves background consistency rather than generally altering image statistics.
>
> **Response 1:** We thank the reviewer for this insightful comment regarding the causal role of the proposed low-frequency enhancement module. In response, we have strengthened our evaluation in two complementary ways. First, beyond semantic similarity metrics (e.g., CLIP_b), we introduce a structural background consistency measure based on open-vocabulary object detection (GroundingDINO), which quantifies background object recall relative to the prompt-conditioned reference generation. The following corresponding description has also been added to **Section 4 (Metrics)**.
>
> In addition, to evaluate structural background consistency beyond semantic similarity, we measure background object preservation using a pretrained open-vocabulary object detector (GroundingDINO [Liu et al., 2023]). For each prompt-conditioned generation, objects detected in the image produced by the original Stable Diffusion (SD) model are treated as reference background semantics. The same detector is applied to the corresponding images generated by the proposed method and competing baselines. Background consistency is quantified using background object recall [Liu et al., 2023; Ghosh et al., 2023]. Let $O_o$ denote the set of background object categories detected in the original SD generation and $O_w$ denote the corresponding set detected in a method-generated image. Background consistency is defined as the proportion of reference background objects that remain detectable:
>
> Object Recall $= \| \( O_o \cap O_w \) \| / \| \( O_o \) \| $
>
> Higher object recall indicates stronger preservation of background scene composition relative to the prompt-conditioned reference generation from the original SD model (please see **the last row of Table 1 (Subsection 4.1)** in the revised manuscript).
>
> Second, to isolate the specific contribution of the low-frequency enhancement mechanism, we perform a controlled sensitivity analysis in which the enhancement strength parameter $ \lambda $ is varied while all other components of the generation pipeline are held fixed. The detailed explanation and results have been added to **Appendix E.4**. As shown in **Table 10**, increasing $ \lambda $ within a moderate range leads to consistent improvements in CLIP$_b$ scores, while CLIP$_f$ and FID remain largely stable. For excessively large values of $ \lambda $, a degradation in perceptual quality is observed due to over-smoothing effects. These results support the interpretation that the proposed low-frequency enhancement mechanism exerts a targeted influence on background scene consistency.
>
> | $\lambda$ | CLIP_b ↑ | CLIP_f ↑ | FID ↓ |
> |-----------|----------|----------|-------|
> | 0 (baseline) | 0.31 | 0.34 | - |
> | 0.10 | 0.19 | 0.30 | 0.62 |
> | 0.20 | 0.23 | 0.30 | 0.61 |
> | 0.30 | 0.27 | 0.31 | 0.60 |
> | 0.35 | 0.31 | 0.32 | 0.58 |
> | 0.45 | 0.29 | 0.27 | 0.68 |
>
> **Table 10:** Effect of low-frequency enhancement strength $\lambda$ on background consistency, semantic alignment, and image quality under foreground–background prompt settings.

---

> ### Author Response · Authors · 2026-03-20
>
> **Comment 2:** The fairness evaluation primarily relies on the WinoBias benchmark and deviation ratio metric, which may not fully capture biases in broader real-world prompts. Including additional datasets or evaluation protocols (e.g., more diverse prompts or demographic attributes) would make the fairness claims more convincing.
>
> **Response 2:** We thank the reviewer for highlighting the importance of evaluating fairness beyond occupation-centric prompts. In response, we expand our fairness evaluation to include a more diverse set of prompts generated by ChatGPT 5.3, inspired by the compositional evaluation philosophy of CEB [1], covering broader real-world scenarios. Furthermore, we extend the demographic analysis by incorporating age (young vs. old) alongside gender. The corresponding discussion and results have been added to **Appendix J**.
>
> The results are reported in **Table 16**. Using the deviation ratio metric, the proposed method consistently maintains balanced generation across these broader prompt categories and demographic dimensions. These additional findings strengthen the evidence that the observed fairness improvements generalize beyond the WinoBias benchmark to more diverse real-world prompting conditions.
>
> To provide qualitative insight, **Figure 16** in the revised manuscript presents representative example generations for the first prompt listed in Table 16, demonstrating balanced outcomes with respect to both gender and age under the proposed method. The results indicate that our approach effectively promotes fair generation while preserving background consistency and overall visual quality.
>
> | Prompts | Gender Self-dis | Gender Ours | Age Self-dis | Age Ours |
> |--------|-----------------|-------------|--------------|----------|
> | a photo of a brain surgeon preparing for a complex operation in a modern hospital | 0.18 | **0.08** | 0.10 | **0.05** |
> | a photo of a technology startup founder presenting a new product on stage | 0.11 | **0.05** | 0.08 | **0.03** |
> | a cinematic shot of a senior executive leading a tense boardroom meeting | 0.21 | **0.14** | 0.26 | **0.16** |
> | a documentary-style image of a renowned physicist giving a public lecture | 0.18 | **0.12** | 0.16 | **0.12** |
> | a construction site worker operating heavy machinery at sunrise | 0.51 | **0.36** | 0.32 | **0.23** |
> | a mechanic repairing a high-performance race car in a professional garage | 0.26 | **0.15** | 0.24 | **0.11** |
> | a hospital caregiver comforting a patient in a quiet recovery room | 0.23 | **0.11** | 0.14 | **0.06** |
> | a primary school classroom teacher guiding students during a reading lesson | 0.16 | **0.07** | 0.21 | **0.13** |
> | a software developer working late at night in a small apartment workspace | 0.19 | **0.09** | 0.22 | **0.14** |
> | a distinguished professor writing notes in a historic university office | 0.16 | **0.08** | 0.25 | **0.12** |
> | **Average** | 0.22 | **0.12** | 0.20 | **0.11** |
>
> **Table 16:** Fairness evaluation on demographically complex and diverse prompts that implicitly induce gender and age bias. Results are measured using the deviation ratio $\Delta$ (lower is better).
>
> [1] Wang, S., Wang, P., Zhou, T., Dong, Y., Tan, Z. and Li, J., CEB: Compositional evaluation benchmark for fairness in large language models, 2024. URL https://arxiv. org/abs/2407.02408.
>
> **Comment 3:** The experiments are limited to U-Net–based diffusion models which are outdated. Since the field is increasingly moving toward transformer-based diffusion models, the authors should discuss limitations more explicitly or provide preliminary experiments demonstrating how the method might extend beyond U-Net architectures.
>
> **Response 3:** We thank the reviewer for this important observation. We agree that the current experimental validation is conducted on U-Net–based diffusion architectures, which, although still widely used in controlled studies on interpretability and fairness, may not fully reflect the recent shift toward transformer-based diffusion models. To address this concern, we have added an explicit statement in **Section 4** clarifying this limitation and better contextualizing the scope of our empirical evaluation.
>
> Furthermore, as noted in the conclusion, extending the proposed framework to vision-transformer–based diffusion architectures (e.g., SD3, PixArt, and related models) constitutes an important direction for future work. Since the proposed method operates through representation-level modulation rather than architectural modification, we believe that the underlying principles remain applicable beyond U-Net backbones.
>
> "We note that our experimental validation focuses on U-Net–based diffusion models; while this enables controlled comparison with prior h-space methods, evaluating the proposed framework on emerging transformer-based diffusion architectures constitutes an important direction for future work."

---

> ### Author Response · Authors · 2026-03-20
>
> **Comment 4:** The concept vector learning procedure involves multiple components (attribute-separation masks, spatial weighting maps, and weighted loss). A more concise algorithmic description summarizing the method would improve readability.
>
> **Response 4:** To improve clarity, we summarize the proposed concept vector learning process as a unified training pipeline that integrates attention-based attribute suppression, spatial modulation of the bottleneck representation, and targeted reconstruction optimization. The overall workflow is illustrated in **Fig. 2, Fig. 8**, and pseudo-code in Appendix C. To further enhance readability, we have added an easy-to-follow high-level overview of the proposed method in **Appendix C**, placed before the pseudo-code:
>
> “Given a target attribute, we first generate a target-conditioned image and its corresponding attribute-attentive heatmap using a frozen diffusion model. These intermediate outputs are used to construct two complementary suppression mechanisms. The first mechanism is the attribute-separation mask applied within the multi-head cross-attention modules of the encoder, which selectively attenuates attention responses associated with the target attribute tokens. The second mechanism is a spatial weighting map derived from the inverted heatmap and encoded into the bottleneck representation to suppress residual attribute traces that may persist after attention masking.
>
> After suppressing attribute information in the encoder output, a learnable concept vector is introduced at the bottleneck layer. The model is then optimized using a spatially weighted reconstruction loss that emphasizes attribute-relevant regions while reducing the influence of spurious contextual features. This sequential procedure ensures that the concept vector becomes the primary carrier of the target attribute information. The combined effect of attention-level suppression, spatial modulation in the latent representation, and region-focused optimization enables interpretable and linearly controllable concept encoding within the diffusion model.”
>
>
> **Comment 5:** The method introduces additional operations (attention masking, heatmap computation, wavelet transforms). Reporting training and inference overhead compared to baseline h-space methods would be helpful to understand practical trade-offs.
>
> **Response 5:** We thank the reviewer for raising this important point regarding the computational overhead of the proposed methodology. To address this concern, we provide a quantitative runtime comparison with baseline h-space methods, separating the computation into three stages corresponding to Algorithm 1 (data generation), Algorithm 2 (learning), and Algorithm 3 (inference). A detailed explanation together with quantitative results has been added to **Appendix F**.
>
>
> Importantly, several components introduced by our method, specifically the generation of attribute-attentive heatmaps and the construction of attribute-separation masks, are executed during a dataset preparation stage prior to optimization. These quantities are computed once and stored, and therefore they do not introduce repeated overhead during the training iterations. As a result, their cost should be interpreted as one-time preprocessing overhead rather than recurring training cost.
>
> To make this distinction explicit, **Table 11 in Appendix F** reports the comparison of the running time for Algorithm 1 (data generation), Algorithm 2 (learning), and Algorithm 3 (inference). The data generation stage includes operations required to prepare the training dataset, while the training stage reflects the actual optimization procedure that is directly comparable across methods. All simulations were conducted on an NVIDIA H100 GPU with 80 GB of memory.
>
> During Algorithm 2, the additional computation introduced by our method is limited to an additional encoder-side pass associated with the heatmap input and lightweight element-wise spatial operations used in the proposed suppression and weighted loss formulations. Since the backbone diffusion model remains frozen, the resulting overhead in the training loop is modest relative to the baseline h-space optimization procedure. In Algorithm 3, the only additional computation introduced by our method is the application of a low-frequency enhancement step, which involves computing the Discrete Wavelet Transform (DWT) of the h-vector.
>
> | Stage | Metric (second) | H-G | Self-dis | Ours |
> |------|------------------|-----|----------|------|
> | Alg. 1: Data Generation | Single training image generation time | 0.96 | 0.96 | 1.48 |
> | Alg. 2: Learning | Time for single training epoch (1k images) | 78 | 47 | 71 |
> | Alg. 3: Inference | Inference time | 3.10 | 2.36 | 3.78 |
>
> **Table 11:** Runtime comparison for Algorithm 1 (data generation), Algorithm 2 (concept vector learning), and Algorithm 3 (inference) across different h-space methods using an NVIDIA H100 GPU with 80 GB memory.

---

> ### Author Response · Authors · 2026-03-20
>
> **Comment 6:** The concept vector manipulation experiments are interesting; further discussion or analysis on how stable and composable these vectors are across prompts and seeds would strengthen the interpretability claims.
>
> **Response 6:** We thank the reviewer for this insightful suggestion regarding the stability and composability of the learned concept vectors. To further strengthen the interpretability claims, we have added an additional analysis evaluating prompt composability and robustness in **Appendix K**. Specifically, we fix a learned concept vector and random seeds, and apply them across a diverse set of human-centric and non-human-centric complex prompts. The corresponding results are presented in **Figure 17 (in the revised manuscript)**. These results demonstrate that the learned concept vectors function as reusable semantic controls that generalize across prompt variations rather than behaving as prompt-specific artifacts.
>
> Furthermore, **Figure 18 (in the revised manuscript)** illustrates the composability of multiple concepts across both human-centric and non-human-centric prompts, highlighting the effectiveness of linear combinations of different learned concept vectors in our method.
>
> **Broader Impact Concerns:** The work aims to improve fairness and safety in text-to-image models, which has positive societal implications. However, the ability to manipulate semantic attributes via concept vectors could potentially be misused to intentionally generate biased or misleading images. In addition, the fairness evaluation is limited to specific datasets and attributes, leaving open questions about broader demographic coverage. A brief discussion of potential misuse and limitations of the fairness evaluation would strengthen the broader impact considerations.
>
> **Response:** We thank the reviewer for highlighting this important consideration. We acknowledge that methods designed to provide explicit control over semantic concept representations may introduce potential dual-use risks. In response, we have expanded **Section 6 (Impact Statement)** to explicitly note that such mechanisms could, in principle, be misused to deliberately inject or amplify biased or sensitive attributes, as discussed below:
>
> "While our work aims to improve fairness and safety in text-to-image models, we acknowledge that explicit control over semantic concept vectors could introduce dual-use considerations. In principle, such mechanisms could be misused to intentionally inject or amplify biased attributes. However, similar controllability already exists in many generative modeling approaches. Our goal is to provide a transparent and interpretable framework for identifying, auditing, and mitigating attribute leakage and bias in diffusion models rather than leaving such associations implicitly embedded in opaque latent representations."

---

### Decision · Action_Editor_QYcC · 2026-04-25

**Recommendation:** Accept as is

**Audience:**

Yes

**Audience Explanation:**

The authors study an important challenge relevant to text2image diffusion models. These models are widely used by the community and the industry. The proposed solution can improve the controllability of these models, which is an ongoing and important issue. I therefore believe that the TMLR audience would be interested in this paper.

**Claims And Evidence:**

Yes

**Claims Explanation:**

The paper focuses on developing a method for improving the controllability (fairness, subject fidelity, and background coherence) of text-to-image diffusion models. The main contributions are frameworks for training and for inference. The problem is well motivated by problems with existing solutions. The reviewers agree that the work is solid and well presented. The authors' results demonstrate consistent improvements across various settings. The reviewers raised a few concerns, primarily around metrics, architectural scope, and causal validation. In my view (and based on the recommendations), the authors properly addressed these issues. Overall, the paper provides practical solutions with strong empirical validation and useful insights, which I believe will be of interest to the ML community. The technical details were validated by all reviewers, so the paper is scientifically sound. Therefore, I recommend acceptance of the paper.